# Wnt/β-catenin signalling is required for pole-specific chromatin remodeling during planarian regeneration

Eudald Pascual-Carreras [1], Marta Marín-Barba[2], Sergio Castillo-Lara[1], Pablo Coronel-Córdoba[1], Marta Silvia Magri[3], Grant N. Wheeler [2], Jose Luis Gómez-Skarmeta[3], Josep F. Abril [1], Emili Saló [1]✉ & Teresa Adell [1]✉

For successful regeneration, the identity of the missing tissue must be specified according to the pre-existing tissue. Planarians are ideal for the study of the mechanisms underlying this process; the same field of cells can regrow a head or a tail according to the missing body part. After amputation, the differential activation of the Wnt/β-catenin signal specifies anterior versus posterior identity. Initially, both *wnt1* and *notum* (Wnt inhibitor) are expressed in all wounds, but 48 hours later they are restricted to posterior or anterior facing wounds, respectively, by an unknown mechanism. Here we show that 12 hours after amputation, the chromatin accessibility of cells in the wound region changes according to the polarity of the pre-existing tissue in a Wnt/β-catenin-dependent manner. Genomic analyses suggest that homeobox transcription factors and chromatin-remodeling proteins are direct Wnt/β-catenin targets, which trigger the expression of posterior effectors. Finally, we identify FoxG as a *wnt1* up-stream regulator, probably via binding to its first intron enhancer region.

During embryonic development, specification of the body axis is one of the earliest events, creating a coordinate system to which to refer when building all organs and tissues. Specification of the first body axis requires the formation of an organizing center or organizer, which refers to a group of cells with the ability to instruct fates and morphogenesis in surrounding cells, giving rise to specific organs and tissues[1–3]. Spemann and Mangold were the first to demonstrate that the dorsal lip of a newt's early gastrula has the ability to generate a fully patterned secondary axis when grafted to the opposite site[4–8]. The homologous organizer is found during gastrulation of all vertebrates, receiving different names, such as the Hensen's node in birds[9] or dorsal shield in fish embryos[10]. The difference between organizers and organizing centers is commonly attributed to their ability to pattern a whole-body axis or just an organ or tissue, respectively. Organizing

centers have been identified in several stages of development, for instance, in the limb bud of tetrapods or the isthmic organizer at the midbrain–hindbrain boundary[1,11,12]. Although organizers are commonly studied in embryos, the first experiment that demonstrated the existence of an organizer was performed in adult *Hydra* by Ethel Browne in 1909. Browne transplanted non-pigmented oral tissue into the body column of a pigmented *Hydra* and observed the induction of a secondary axis that was composed of the cells of the host[13]. More than a century later, the existence of organizing regions in adult tissues, i.e., during regeneration, which also requires re-patterning of tissues, has not been well studied. In this study we investigated the process of regeneration of planarians, flatworms that can regenerate any missing structure, even the head, in a few days. Thus, they are whole-body regenerating animals, which need to re-pattern the body

[1]Department of Genetics, Microbiology and Statistics, Universitat de Barcelona (UB) & Institute of Biomedicine of Universitat de Barcelona (IBUB), Barcelona, Spain. [2]School of Biological Sciences, University of East Anglia, Norwich Research Park, Norwich, UK. [3]Centro Andaluz de Biología del Desarollo (CABD), Universidad Pablo de Olavide, Sevilla, Spain. ✉e-mail: esalo@ub.edu; tadellc@ub.edu

axes to regenerate the proper missing structures according to the pre-existing polarity.

The plasticity of planarians is based on the presence of a population of adult pluripotent stem cells (called neoblasts)[14,15], in addition to the continuous activation of the signaling pathways that instruct the fate of these stem cells and their progeny. Several studies demonstrate that muscle cells surrounding the planarian body are the source of Positional Control Genes (PCGs), which are secreted factors that confer axial identity to the remainder of the cells[16–20]. A subset of these muscular cells located in the most anterior (tip of the head) and the most posterior (tip of the tail) ends of the planarian body act as organizers[21]. The anterior and the posterior tip express the PCGs *notum* (a Wnt/β-catenin or cWnt pathway inhibitor) and *wnt1* (a cWnt pathway activator), respectively. Inhibition of these genes produces a shift in the polarity, originating two-tailed or two-headed planarians after silencing *notum* or *wnt1*, respectively[22–24]. Thus, in planarians, as described in several embryonic models, the cWnt pathway specifies the anterior–posterior axis[25–30]. Importantly, during the first hours of regeneration, both *notum* and *wnt1* are expressed in differentiated cells of any wound, and it is not until 36–48 h of regeneration that they are restricted to their respective tip[22,24,31], forming the anterior and the posterior organizing regions. It is known that this late localized expression of *notum* and *wnt1* depends on the proliferation of stem cells, and that it requires the expression of specific transcription factors and kinases, such as *foxD*, *zicA*, *prep*, or *pbx* for anterior tips[32–36] and *islet*, *pitx*, *mob, striatin*, and *teashirt* (*tsh*) for posterior tips[37–41]. However, the triggering of the early expression of *notum* and *wnt1*, which does not depend on stem cell proliferation, is not understood.

Since gene expression is defined by the epigenome, early changes in the chromatin accessibility in the wound region must occur, in order to reprogram the fate of the preexistent tissue. Thus, in this study, we undertook a genomic approach to analyze the formation of the posterior organizer during the early stages of planarian regeneration. Through ATAC-sequencing and ChIPmentation techniques, we uncovered *cis*-regulatory elements (CREs) of the *Schmidtea mediterranea* genome[42] and analyzed their accessibility in wild type (WT), *notum*, and *wnt1* (RNAi) regenerating wounds. Our results show that at 12 hours of regeneration (hR), anterior wounds of *notum* (RNAi) animals resemble WT posterior wounds, and posterior wounds of *wnt1* (RNAi) animals resemble WT anterior wounds. Thus, during the first hours after amputation, before the expression of any anterior or posterior marker, the chromatin accessibility of the cells localized in the wound region has changed according to the polarity of the pre-existing tissue. Analyzing the DNA binding motifs upstream of genes downregulated after *wnt1* (RNAi), we found an enrichment of Homeobox transcription factors (TFs), suggesting that these are the genes directly regulated by the cWnt pathway (Wnt1/β-catenin-1) and responsible for triggering the posterior program. Moreover, thanks to the annotation of CREs, we identified two putative enhancer regions located in the first intron of *wnt1* containing a FoxG binding site. Silencing of *foxG* inhibits the early and late phases of *wnt1* expression, but not *notum*, and phenocopies the *wnt1* (RNAi) phenotype. This result suggests that FoxG directly regulates the early expression of *wnt1* in any wound and is a key factor in triggering the formation of the posterior organizing center, and thus specifying posterior versus anterior identity.

In this study we annotate CREs (promoters and enhancers) functional during planarian regeneration and specific of anterior-vs-posterior pole identity acquisition. An open platform to query and interpret all transcriptomic and genomic results obtained has been created (https://compgen.bio.ub.edu/PlanNET/planexp[43] and https://compgen.bio.ub.edu/jbrowse/). An additional web-tool has been developed to search for transcription factor binding sites and to explore the predicted regulatory elements (https://compgen.bio.ub.edu/PlanNET/tf_tools).

## Results

### The chromatin of cells in the wound region remodels according to the polarity of the pre-existing tissue

To identify CREs that after amputation could specify anterior or posterior identity, we performed ATAC-sequencing and ChIPmentation of anterior and posterior wounds 12 h after post-pharyngeal amputation of *S. mediterranea* (Fig. 1a). At this regeneration time point, the early expression of the cWnt elements *notum* and *wnt1* is first detected, although it is still not polarized[22,24,31]. The comparison of the results obtained when analyzing anterior versus posterior wounds allowed us to identify ATAC-seq peaks corresponding to accessible chromatin regions (ACRs) specific for each pole (DiffBind, FDR < 0.05, fc > 2, see Computational Supplementary Methods—CSM from now on–, sections 4 and 5). We found 611 anterior specific ACRs and 2484 posterior specific ACRs. Comparing these ACRs with ChIPmentation of samples also corresponding to 12 hR anterior and posterior wounds using the H3K27ac antibody, which allows the identification of active enhancers[44,45], enabled us to identify a list of 555 anterior putative active enhancers and 1869 posterior putative active enhancers (Fig. 1a and Supplementary Data 1). Using HOMER, we studied the presence of TF binding motifs in those anterior and posterior putative active enhancers (CSM, sections 6 and 7). Binding motifs for PITX, a homeobox TF that controls Wnt-dependent tail formation in planarians[38], were one of the most represented in posterior putative enhancers. Binding sites for LHX, PBX and CUX, which are anterior and neural regulators in planarians[35,46,47], were the most represented in anterior putative enhancers (Supplementary Fig. 1, Supplementary Data 2). Thus, the predicted TFs binding sites presence validates the specificity of the identified CREs with respect to the anterior–posterior fate.

Silencing of *notum* or *wnt1* during planarian regeneration produces a shift in polarity, giving rise to anterior tails in *notum* (RNAi) animals[22,48] and posterior heads after *wnt1* (RNAi)[23,24,49]. With the aim of analyzing the chromatin changes occurring during anterior and posterior specification, we performed ATAC-seq of *notum* (RNAi) anterior wounds and *wnt1* (RNAi) posterior wounds, both at 12 hR. We analyzed the state of the anterior and posterior putative active enhancers previously found to be specifically open in the anterior or posterior in these RNAi samples. The result shows that in *notum* (RNAi) anterior wounds, only 12.3% of the anterior putative active enhancers were open, whereas the remainder were closed or reduced their accessibility (Fig. 1b, Supplementary Figs. 2, 3 and 4, and Supplementary Data 1). Moreover, 87.7% of the posterior putative active enhancers were accessible in *notum* (RNAi) anterior wounds. In *wnt1* (RNAi) posterior wounds, only 24.5% of the posterior putative active enhancers were open and the remainder were closed or decreased their accessibility. Furthermore, 31.4% of the anterior putative active enhancers appeared to be open in *wnt1* (RNAi) posterior wounds and 9.5% became more accessible (Fig. 1b, Supplementary Figs. 2, 3 and 4, and Supplementary Data 1).

In summary, we found specific CREs and potential associated TF in anterior or posterior wounds. The accessibility of these putative enhancer regions changes as soon as 12 h after amputation in *notum* or *wnt1* (RNAi) anterior or posterior wounds, respectively. These results indicate that (1) inhibition of the key elements of the anterior and posterior organizers, *notum* and *wnt1*, respectively, produces a very early change of the chromatin structure, suggesting that both elements trigger the specific anterior or posterior program via the regulation of chromatin remodelers; and (2) the chromatin structure of cells in the wound is remodeled according to the polarity of the pre-existing tissue, which occurs a few hours after amputation (<12

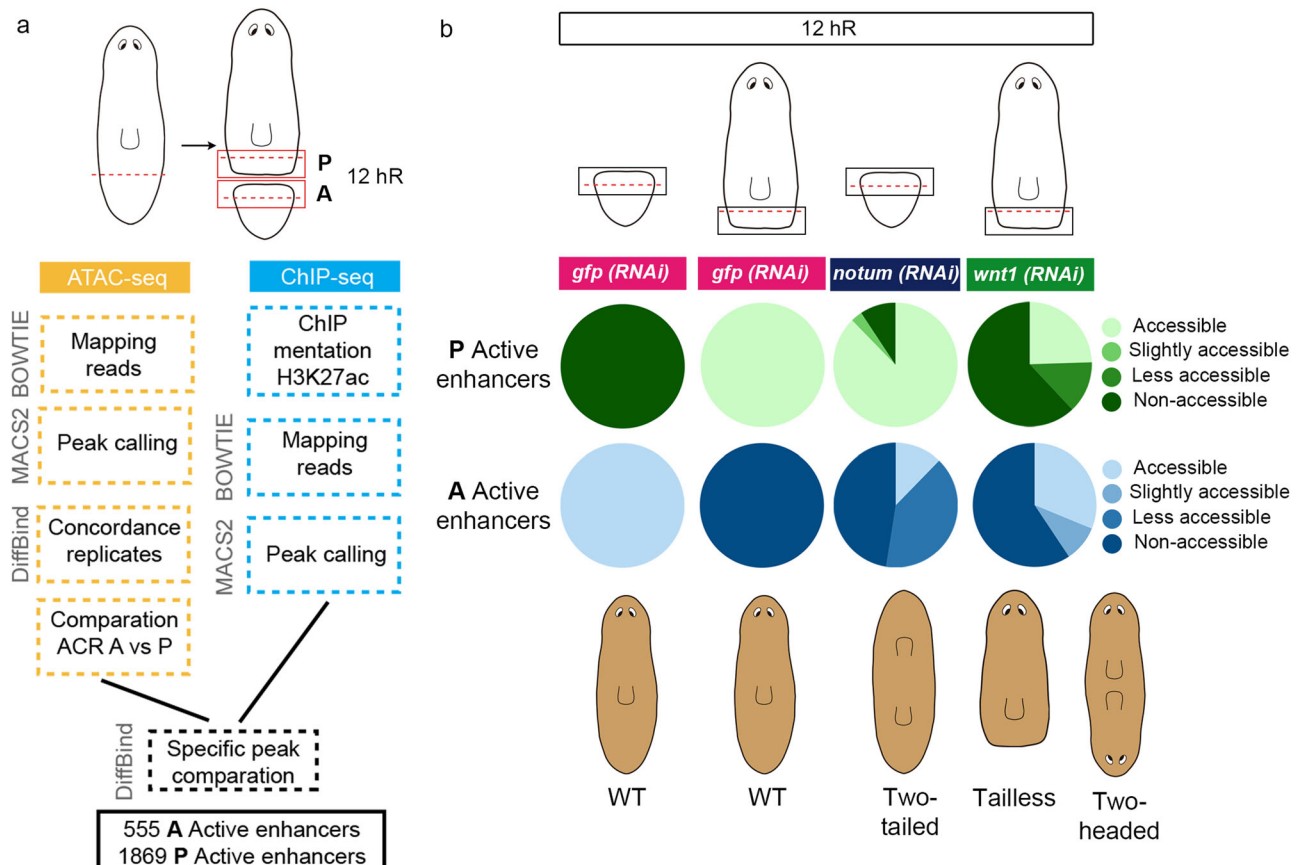

**Fig. 1 | The chromatin of cells in the wound remodels according to the polarity of the pre-existing tissue. a** Workflow to identify putative anterior and posterior specific enhancers at 12 hR. Next to each step the program used is indicated. **b** Accessibility changes of the putative anterior and posterior specific active enhancers after *notum* and *wnt1* inhibition at 12 hR are represented in percentages in pie charts. Schematic illustration shows the representative phenotypes observed after each gene inhibition. hR, hour of regeneration. All the data is provided in Supplementary Data 1.

hR), before the first anterior or posterior markers appear (around 48 hR)[50].

### Homeobox TFs motifs are found to be enriched in *Cis*-Regulatory Elements of *wnt1* (RNAi) downregulated genes

To identify CREs that could be regulated by the cWnt pathway during posterior regeneration, we performed an RNA-seq of controls and *wnt1* (RNAi) posterior wounds (0–72 hR) to find the genes downregulated after cWnt pathway inhibition (Fig. 2a). We performed differential expression analysis (padj <0,05, fc ± 0,5) at each time point (Supplementary Data 3). A total of 2129 genes were found to be differentially expressed at any time point; among these, 712 genes were downregulated in *wnt1* (RNAi) planarians with respect to controls (Fig. 2b, Supplementary Data 3 and https://compgen.bio.ub.edu/PlanNET/planexp). As expected, known elements involved in posterior identity specification were found among the *wnt1* (RNAi) downregulated genes (Fig. 2b and Supplementary Data 3): the posterior Homeobox genes (*Smed-hox4b*, *Smed-post-2c Smed-post-2b*, *Smed-lox5a*, and *Smed-lox5b*)[26,29,51–53]; the posterior Wnt11 (*wnt11-1* and *wnt11-2*)[23,27,31]; the posterior Frizzled *fzd4-1*[30,54], *axinB*[26], *tsh*[39,40], and sp5, a TF recently found to mediate the evolutionary conserved role of cWnt in axial specification[29].

To analyze the CREs of the *wnt1* (RNAi) downregulated genes, we first identified the CREs found in 12 hR planarian wounds using the previous ATAC-seq and ChIPmentation samples, together with ATAC-seq analysis of un-wounded planarians (0 hR) and anterior and posterior 48 hR regenerating wounds (https://compgen.bio.ub.edu/

jbrowse/). We classified the CREs found in putative promoters or enhancers according to their position with respect to the Transcriptional Start Site (TSS) (Fig. 2a) as described in Material and Methods section. The putative promoters were classified as Core Promoters (CP) and Proximal Promoters (PP), and we identified 2594 and 1549 of each, respectively. The putative enhancers were classified as First Intron (FI), Proximal (Pro), and Distal (Dis), and we identified 3157, 19,610, and 28,720 of each, respectively (see CSM, section 7).

Using HOMER, we analyzed the presence of TF binding motifs in the CREs of the 712 genes downregulated in *wnt1* (RNAi) wounds. The result shows that the motifs found in the putative enhancer regions of a higher percentage of *wnt1* (RNAi) downregulated genes were Homeobox (Fig. 2c, Supplementary Data 4). Considering that posterior identity is specified by the cWnt signaling (Wnt1/β-catenin-1/TCF axis)[55], we then searched for the CREs containing a TCF binding site. We found 167 genes containing a TCF binding site in the putative enhancer region, 17 of which also showed a TCF motif in the promoter (Supplementary Data 4). Among these, we found the genes already known to be involved in posterior specification: posterior Hox genes (*lox5b, hox4b, post2c*)[29,53], sp5[29], *axinB*[26], and *tsh*[39,40], indicating that they are direct targets of the Wnt1/β-catenin-1/TCF signaling. We also combined our RNA-seq data with the RNA-seq of *β-catenin-1* (RNAi) animals already reported[29]. This strategy resulted in 42 genes (Supplementary Data 3 and 4), which included most of the posterior genes already found to possess a TCF binding site, further supporting the direct role of these candidates in specifying posterior via cWnt signaling.

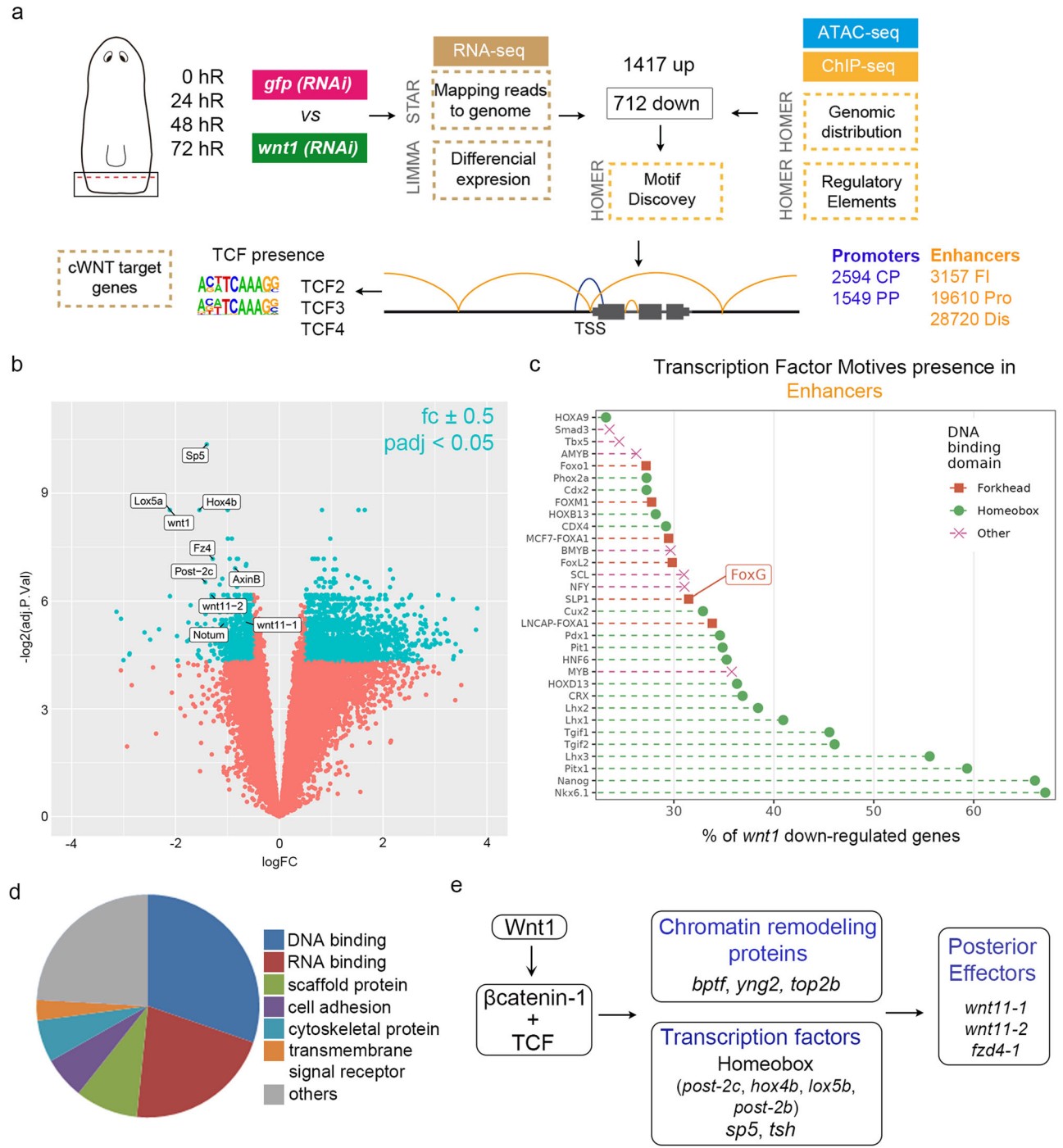

**Fig. 2 | Homeobox TFs motifs are found enriched in cis-regulatory elements of *wnt1* (RNAi) downregulated genes. a** Workflow to identify differentially expressed genes, *cis*-regulatory elements (CRE) and transcription factors (TFs) associated with *wnt1* (RNAi). Next to each workflow the program used is indicated. Motif discovery for TCF binding sites was specifically performed in downregulated *wnt1* (RNAi) genes. **b** Volcano plot of 72 hR shows the (694) down- and (1409) upregulated genes after *wnt1* inhibition, which present fold change (fc) ± 0.5 and *p*-value adjusted (padj) <0.05 (FDR on lima-voom empirical Bayes moderated t-test).

Significant genes are colored light blue, and not significant are colored light red. Some significant genes are labeled. Data provided in Supplementary Data 3. **c** TF motifs presence on the putative enhancer regions of *wnt1* (RNAi) downregulated genes, showing a higher representation of Homeobox. Data provided in Supplementary Data 4. **d** Gene Ontology (molecular function) analysis of the *wnt1* (RNAi) downregulated genes containing TCF binding sites (http://pantherdb.org/). Data provided in Supplementary Data 5. **e** Schematic illustration of the proposed genetic program activated by *wnt1* in posterior wounds.

Several genes downregulated in *wnt1* (RNAi) wounds showed a TCF binding site. However, the most represented motifs found in *wnt1* (RNAi) downregulated genes are not TCFs but Homeobox. These results suggest that the Wnt1/β-catenin-1/TCF signaling could directly activate the expression of TFs, which in turn would activate the effectors of the posterior fate. In agreement with this hypothesis, Gene

Ontology enrichment analysis of the genes downregulated after *wnt1* (RNAi) with TCF binding site in their putative promoter or enhancer regions showed a significant enrichment of DNA binding proteins/regulators of transcription and specifically of Homeobox TFs (Fig. 2d, Supplementary Data 2–5, Supplementary Fig. 5). Among the *wnt1* (RNAi) downregulated genes containing TCF motifs, we also found

chromatin-remodeling proteins, such as BPTF, a nucleosome-remodeling factor[56], and TOP2B[57] (Supplementary Data 2–5, Fig. 2d). This result agrees with the rapid changes in chromatin accessibility that we observed at 12 h of regeneration. Interestingly, the TFs required for regeneration of longitudinal and circular muscular fibers, *myoD* and *nkx-1*, respectively[19], which are the source of the PCGs, are also found among the *wnt1* (RNAi) downregulated genes containing a TCF binding motif. A new web tool was developed to search for transcription factor binding sites and to explore the predicted regulatory elements (https://compgen.bio.ub.edu/PlanNET/tf_tools).

Overall, we identified new genes, TFs and CREs participating in the specification of posterior identity; some of these were already known to specify posterior, validating our strategy, and many of these are new elements of the Wnt1 gene regulatory network. Our data suggests that the Wnt1/β-catenin-1/TCF signaling directly activates chromatin-remodeling complexes and Homeobox genes, among other TFs (*sp5*, *tsh*), which in turn regulate the activation of the effectors of posterior specification (Fig. 2e).

### Cis-regulatory element (CRE) found in *wnt1* first intron contain FoxG motives

Taking advantage of the previous analysis, which allowed the mapping of CREs in the *S. mediterranea* genome[42], we sought to investigate the presence of CREs in the *wnt1* locus to understand the regulation of its expression. We found different evidence that the first intron of *wnt1* presented two putative enhancer regions, which were named enhancer 1 (E1) and enhancer 2 (E2) according to their distance to the TSS (Fig. 3a and Supplementary Fig. 6). We found that both regions show: (i) a nucleosome free region (ATAC-seq peak) and (ii) that this region is correlated with histone modifications related to enhancer activity (H3K27ac ChIPmentation; Fig. 3a). Interestingly, E2 was recently identified in an independent study[58]. While the E1 seems to be specifically accessible at 12 hR, the E2 is accessible prior to amputation and during the regeneration process (Supplementary Fig. 7). Both putative enhancer regions were located less than 3 kb from the *wnt1* promoter, suggesting that they could regulate its expression[59]. Through motif discovery, we analyzed the presence of TF binding sites in both regions and observed the presence of FoxG binding sites (Fig. 3a, Supplementary Fig. 7, Supplementary Data 6), which is also a motif present in a high percentage of CREs of the 712 genes downregulated in *wnt1* (RNAi) wounds (Fig. 2c).

To assess the biological significance of these putative enhancer regions, we looked for evidence showing their evolutionary conservation. Interestingly, we found that, although *wnt1* genes present a variable number of introns, the first intron, which contains the FoxG binding sites, maintains a conserved position in all the genomes analyzed (Fig. 3b). Furthermore, the analysis of the first intron of the different species reported evidence of active enhancer features, such as: existence of accessible chromatin regions (i.e. ATAC-seq, DNase-seq), histone modifications associated with enhancer activity (i.e. H3K27ac, H3Kme1/2/3), and the presence of FoxG binding site motifs (Fig. 3c and Supplementary Data 7). Importantly, a ChIPmentation analysis of *Drosophila melanogaster* using the FoxG antibody demonstrates the binding of Dm-FoxG (SLP1) to the first intron of *Dm-wnt1* (*wingless*)[60,61].

Thus, the finding of enhancer activity in *S. mediterranea wnt1* first intron and its possible evolutionary conservation, supports the role of FoxG as a direct *wnt1* regulator.

### FoxG regulates *wnt1* expression and specifies posterior identity

To further investigate the potential role of FoxG regulating *wnt1* expression in planarians, we inhibited it by RNAi in regenerating animals (Fig. 4a). Whole mount in situ hybridization (WISH) of *wnt1* in *foxG* (RNAi) animals demonstrated that it was absent in both at 12 hR and 3 dR posterior wounds, indicating that *foxG* is required for both the early (stem cell independent) and the late (stem cell dependent) phase of *wnt1* expression (Fig. 4b). *foxG* was also necessary for the expression of *wnt1* in the anterior 12 hR wounds (Fig. 4b). Furthermore, inhibition of *foxG* in intact animals also led to the disappearance of *wnt1* expression (Supplementary Fig. 8a). However, the early and late phases of *notum* expression were not affected (Supplementary Fig. 8b).

In agreement with a direct role of *foxG* in regulating *wnt1* expression, WISH analysis showed that *foxG* is expressed in the posterior dorsal midline (Fig. 4c), and in scattered cells in both anterior and posterior-facing wounds at 24 h of regeneration (hR) (Fig. 4d), as described with *wnt1* expression[23,24] (Fig. 4d). *foxG* is also expressed in cells along the D/V margin, in scattered cells in the dorsal and ventral part of the animals and in the central nervous system, as shown in 72 h regenerating animals (Supplementary Fig. 8c). Single Cell Sequencing (SC-seq) database analysis indicates that *foxG* cells could be muscular and neuronal[62] (Fig. 4c and Supplementary Fig. 9a). Of note, *foxG* is one of the top specific genes found in muscular *wnt1*+ cells of the posterior midline (*wnt1*+ and *collagen*+) in intact animals and in posterior regenerating blastemas at 72 hR[43,63] (Fig. 4c), further suggesting its role in the specification of the posterior organizer through regulating *wnt1* expression.

If *foxG* is required for *wnt1* expression, then regenerating *foxG* (RNAi) animals should show a phenotype related to the malfunction of the posterior organizer. Accordingly, we found that 70% (15/20 and 40/56) regenerating *foxG* (RNAi) animals presented a rounded posterior blastema (Fig. 5a, b), resembling the tailless phenotype obtained in the mild *wnt1* (RNAi) phenotype[23,24,31]. Analysis of the central nervous system and the digestive system by anti-arrestin (3C11) and anti-βcatenin-2 immunohistochemistry, respectively, demonstrated that 70% (15/21) of these animals are tailless (Fig. 5b and Supplementary Fig. 9b). They show a fusion of the posterior nerve cords and intestine branches in a U shape, as has been described after inhibition of other key posterior genes, such as *wnt1*[23,49], *wnt11-2*[31,49], *islet*[37,38] or *pitx*[38]. Furthermore, WISH and qRT-PCR with posterior markers, which we demonstrated in the previous section are cWnt target genes (*fz4*[54], *post2d*[29], sp5[29] and *hox4b*), indicated that they are downregulated in posterior *foxG* (RNAi) regenerating blastemas at 3 dR (Fig. 5b and Supplementary Fig. 9d). Interestingly, after increasing the concentration of *foxG* dsRNA injected (see "Methods"), 2 out of 20 regenerating animals (two independent experiments) showed a two-headed phenotype (Fig. 5a, c), shifting the polarity from posterior to anterior, as observed after *wnt1* or *β-catenin-1* inhibition[26,30].

To further investigate the potential role of *foxG* regulating other genes involved in the regeneration process, we sought the presence of FOXG binding site motifs in wound-induced genes[50], finding that 38 out of 128 wound-induced genes presented a FOXG binding site motif on their CREs (Supplementary Data 8), such as: *egrl-1*[64], sp5[29] and *jun-1*[65]. We also studied the presence of FOXG motifs in the putative anterior and posterior CREs annotated in this study (Supplementary Data 1). The gene ontology (GO) analysis of the associated genes suggests that FoxG regulates different process in anterior and posterior wounds (Supplementary Fig. 10 and Supplementary Data 9).

These results demonstrate that *foxG* is a new element of the posterior organizer. Our data indicates that *foxG* is upstream of *wnt1* because inhibition of *foxG* suppresses *wnt1* expression in all stages and tissue regions, and *wnt1* (RNAi) animals do not show a decrease in *foxG* expression (Supplementary Data 3 and Supplementary Fig. 9c).

## Discussion

The plasticity of planarians is providing insightful data about the mechanism underlying regeneration. Several studies have demonstrated that the anterior and posterior tips of planarians function as organizers (reviewed in ref. [21]), a term that has been traditionally used in the field of embryonic development. The finding of adult organizers

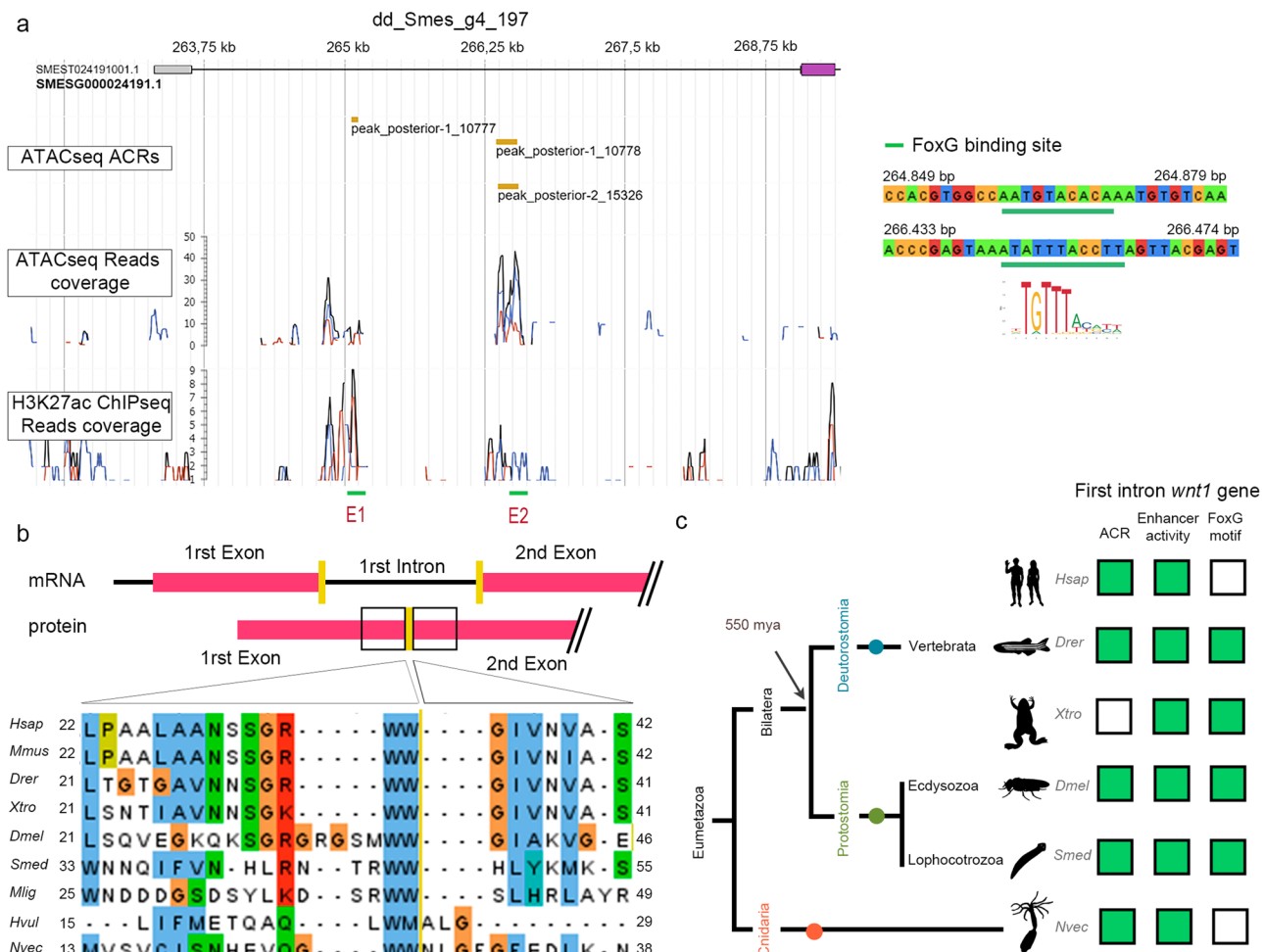

**Fig. 3 | FoxG could bind to a cis-regulatory element found in *wnt1* first intron.**
**a** Schematic illustration of *Smed-wnt1* gene locus, indicating exons (violet boxes) linked by introns (lines). Genome Browser screenshot showing the ATAC-seq Accessible Chromatin Regions (ACRs), ATAC-seq and H3K27ac ChIP-seq profiles of the putative enhancer regions. Putative Enhancer 1 (E1) and Enhancer 2 (E2) present a FoxG motif (SLP1) (green line). The ATAC-seq peaks corresponding to E1 and the E2 are indicated. **b** Alignment of WNT1 amino acid sequences show the conservation of the intron 1 position. Yellow line shows the separation between the first and the second exon. **c** Schematic summary of accessible chromatin regions (ACRs), enhancer activity and FoxG motif evidence in the first intron of *wnt1* genes in

different eumetazoan species. Green box indicates evidence and white box indicates no available data. Data is provided in Supplementary Data 7. Species used: *Homo sapiens* (*Hsap*), *Mus musculus* (*Mmus*), *Danio rerio* (*Drer*), *Xenopus tropicalis* (*Xtro*), *Drosophila melanogaster* (*Dmel*), *Schmidtea mediterranea* (*Smed*), *Macrostumum ligano* (*Mlig*), *Hydra vulgaris* (*Hvul*), and *Nematostella vectensis* (*Nvec*). Silhouettes are from https://beta.phylopic.org. *Xenopus* silhouette is from Sarah Werning and the licence is https://creativecommons.org/licenses/by/3.0/. *Schmidthea* silhouette is from Noah Schlottman and the license is https://creativecommons.org/licenses/by/3.0/.

in other regenerating animals, such as *Hydra*, zebrafish, or *Xenopus* tadpoles, supports the idea that the formation of organizers could be a general mechanism that confers regenerative properties[66–68]. There are common features in the reported examples: (1) the cells that function as organizers are non-proliferative and are located in the periphery of the early blastema, and (2) the organizing activity relies on the cWnt signaling[66,67,69]. These properties are also accomplished by planarian organizers. A difference between planarians and other bilaterian models of regeneration is that planarians can completely regenerate a new axis from both ends, anterior and posterior, independently of the fragment amputated. The plasticity of the model, in addition to the use of genomic and transcriptomic high throughput techniques, has allowed us to compare the genomic changes occurring during anterior or posterior specification in the same field of original cells.

Our data indicates that the establishment of the appropriate identity in a planarian wound could follow a three-step model (Fig. 6). (1) Remodeling of the chromatin, which must occur very early after a cut, even before the appearance of any anterior or posterior marker. We demonstrate that the activity of the cWnt signaling is fundamental

for this remodeling. The chromatin changes in *notum* (RNAi) wounds with respect to WT are stronger than those in *wnt1* (RNAi). However, we cannot conclude that *notum* has a more determinant or earlier effect than *wnt1*, because we only have analyzed one time point, 12 h after the cut. Furthermore, our *wnt1* (RNAi) animals have a milder phenotype (tailless) than *notum* (RNAi) animals (two-tailed). (2) Remodeling of the chromatin could allow the expression of TFs, such as the Hox genes, whose transcription has shown to be dependent on extensive chromatin remodeling in other models[70]. These two steps appear to be directly regulated by the cWnt signaling, because among the *wnt1* (RNAi) downregulated genes showing TCF binding sites in their putative enhancer/promoter regions, we found an enrichment of DNA binding proteins, including chromatin-remodeling proteins and TFs. (3) The third step consists in the activation of the Posterior Effectors (*wnt11-2*, *wnt11-1* or *fzd4-1*…), which are required for differentiation and morphogenesis of tail structures, thanks to the TFs active in step 2.

The early change in the genomic landscape found in each regenerating tip, in addition to the finding of several chromatin-remodeling

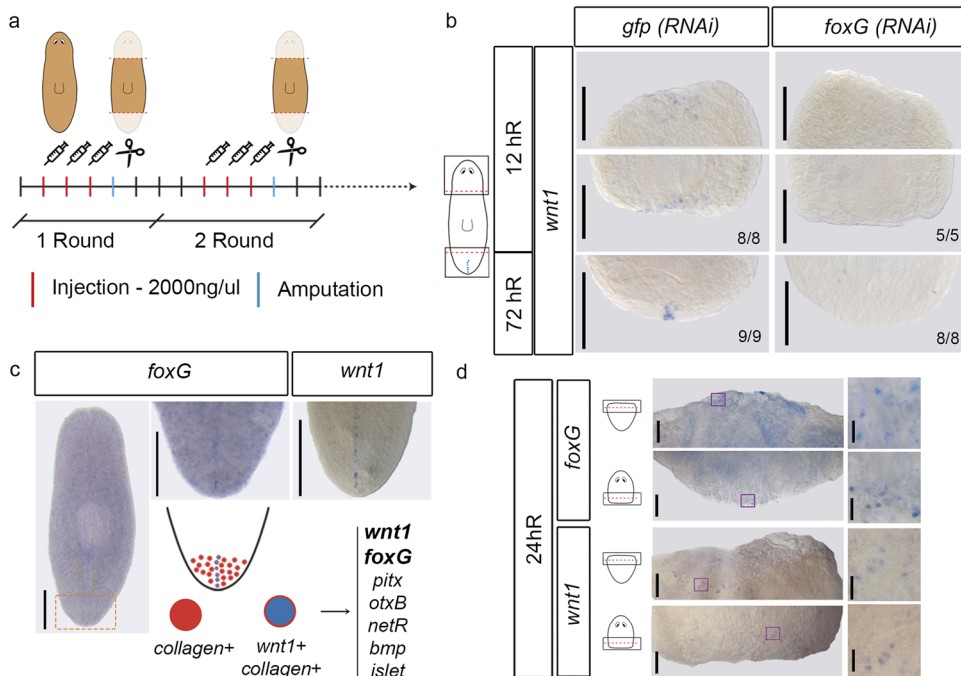

**Fig. 4 | *foxG* is expressed in regenerating blastemas and regulates *wnt1* expression. a** Schematic illustration indicating the *foxG* (RNAi) procedure. **b** WISH of *wnt1* in *foxG* (RNAi) animals demonstrate its absence in both 12 and 72 hR wounds. Schematic illustration of *wnt1* in intact animals with the analyzed zones (squares) added. Data representative of two independent experiments. **c** WISH of *foxG* in intact animals shows its expression in the posterior midline, similar to *wnt1* expression. Data representative of one experiment. Orange dashed lines show the magnified area. Single cell analysis performed by[63] showed six genes (top 16%) over

represented in posterior organizing *wnt1*+ cells. Among these, *foxG* was found. **d** WISH of *wnt1* and *foxG* in regenerating anterior and posterior wounds at 24 hR showing a salt-and-pepper pattern (left panel). A magnification was taken from each blastema (right panel). Purple boxes show the magnified area. Schematic illustration of regenerating animals was added. Data representative of one experiment. Scale bar: 50 μm and 10 μm for left and right panel in b, respectively; 100 μm in c; 200 μm in d.

proteins downregulated in *wnt1* (RNAi) genes showing a TCF binding site, indicates that in planarians the Wnt/β-catenin pathway specifies cell fate via regulating chromatin structure and reprogramming as described in other contexts[71]. It is important to note that the early expression of *notum* and *wnt1* is stem cell independent[24], supporting the essential role that reprogramming could have at this early stage.

Furthermore, our data restricts the timing when this chromatin remodeling occurs, which must be earlier than 12 h after the cut. Thus, a novelty of the proposed three-step model is that, in contrast to the results found in previous transcriptomic analysis in planarians, in which injury-specific transcriptional responses emerged 30 h after injury[50], we observed that changes occurring in the chromatin of cells in each wound are wound-specific and occur a few hours after the cut. These rapid changes at genomic level were visualized due to the genomic analysis restricted to the cells in the wound region and at a very early time point, significantly before the appearance of any polarity signaling. Future genomic analysis at single cell level will allow visualization of the genomic changes occurring in specific cell types of the wound region.

Our RNA-seq analysis agrees with previous transcriptomic studies, because we found that inhibition of *wnt1* leads to the deregulation of a large number of genes at late timepoints (48–72 h), corresponding to the process previously known as injury-specific transcriptional response[50]. However, the second novelty of our three-step model is that through the identification of CREs in the *S. mediterranea* genome, we observed that the motif present in a higher percentage of *wnt1* (RNAi) downregulated genes was the Homeobox binding site and that a small number of them, mainly corresponding to DNA binding proteins, contained TCF motifs. This result suggests that TFs and mainly Homeobox genes are direct targets of WNT1 that afterwards will activate the transcription of the posterior effectors. The Hox genes *post2c*,

*lox5a/b*, and *hox4b* are specifically expressed in posterior, although regenerative and segmentation defects have only been seen after *lox5a* and *lox5b* (RNAi)[17,72], respectively. Not only in planarians, but also in other whole-body regenerating animals, such as acoels and *Hydra,* Hox genes and *sp5*, which also shows a TCF motif in its CRE, have a role in axis establishment, suggesting that a conserved set of cWnt targets mediate the patterning of the primary body axis[29,69].

According to our data, *wnt11-1* and *wnt11-2*, which are required to regenerate a proper tail but whose inhibition does not produce a shift in polarity[23,24,27], are downregulated at a late stage in *wnt1* (RNAi) animals, and do not show a TCF binding motif, suggesting that they could be part of the posterior effectors. Supporting the late role of *wnt11-1* and *wnt11-2*, their silencing inhibits the late *wnt1* expression but not the early expression[49]. The same situation occurs with the posterior WNT receptor, *fzd4-1*. In this case, it could be that the expression of *fzd4-1* is mediated by the early *wnt1*+ cells, as recently proposed[55], forming the posterior organizer, in accordance with the idea that organizers evocate the surrounding tissue; a first organizer action would be to prepare the tissue to make it competent to itself[2].

The fundamental role of the Notum-Wnt1 antagonism in establishing the identity of a wound has been widely demonstrated through functional and expression analyses[22–24,29,49,50,73,74]. The proposed three-step model assumes this antagonism and presupposes that remodeling of the chromatin is different in anterior and posterior wounds because *notum* is expressed at higher levels in anterior and *wnt1* is expressed at higher levels in posterior. Genes required for the late expression of *notum* and *wnt1*, localized in the midline, have been identified. *foxD* and *zicA* (RNAi) animals do not show the late expression of *notum* and do not regenerate a proper head[32–34]; *islet* and *pitx* (RNAi) animals do not show the late expression of *wnt1* and are tailless[37,38,46]. Little was known about the regulation of the expression

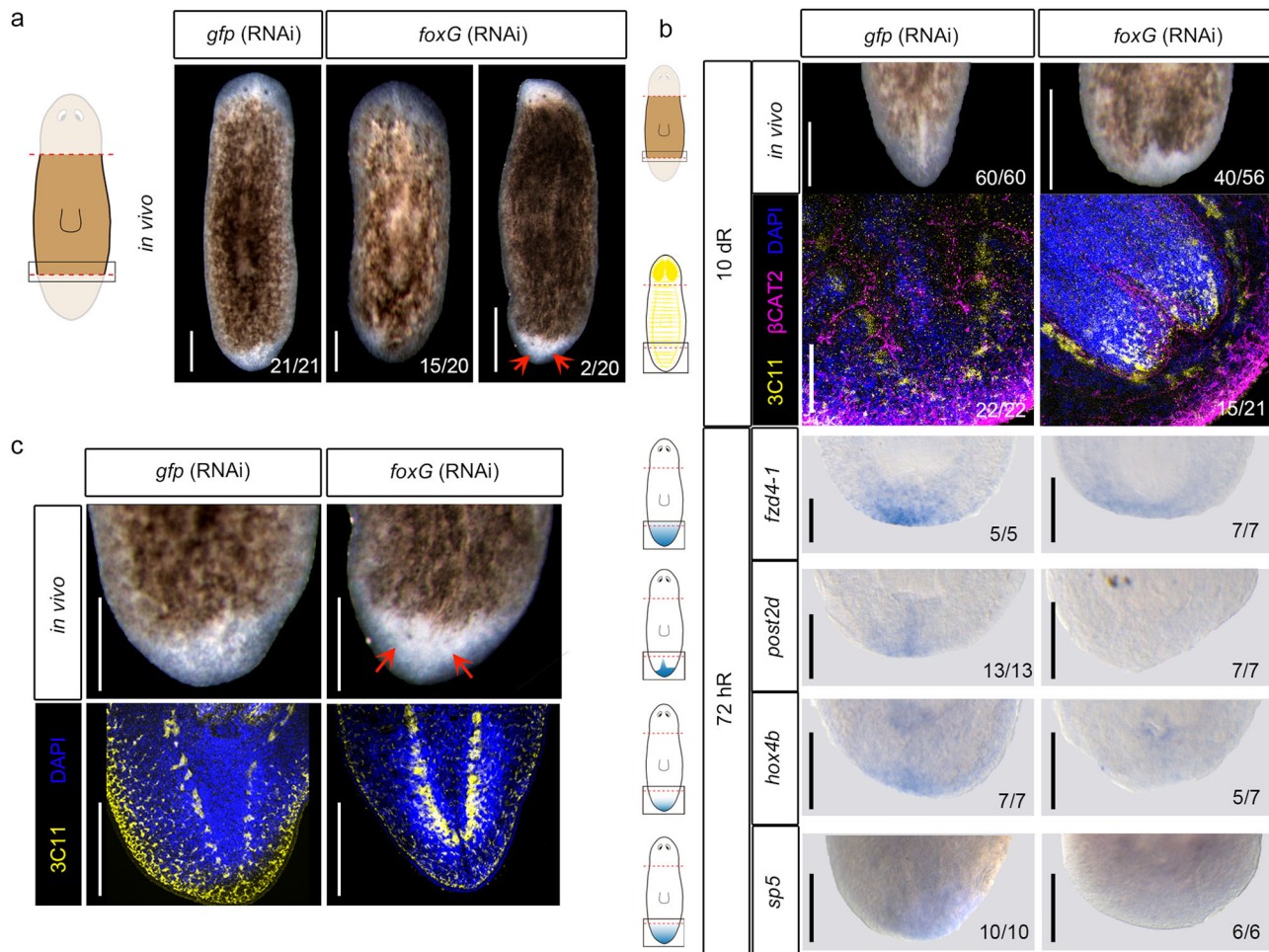

**Fig. 5 | *foxG* (RNAi) phenocopies *wnt1* inhibition. a** In vivo phenotypes after *foxG* (RNAi). Data representative of two independent experiments. **b** Immunostaining using α-SYNAPSIN (3C11) (neural system) and α-βCAT2 (β-catenin-2, digestive system) reveals rounded ventral nerve cords in *foxG* (RNAi) (tailless phenotype). Nuclei are stained in DAPI. WISH of posterior markers in regenerating *foxG* (RNAi) animals demonstrated a reduced expression. Data representative of two independent experiments. Schematic illustrations of posterior markers were added. **c** Immunostaining using α-SYNAPSIN (3C11) (neural system) reveals a posterior brain in the *foxG* (RNAi) two-headed animals. Data representative of two independent experiments. Nuclei are stained in DAPI. Posterior eyes are indicated with an orange arrow in a and c. Scale bar: 100 μm in a, immunostaining in b and c; 200 μm in WISH in b.

of *notum* and *wnt1* in disperse cells of early wounds. *Equinox*, an extracellular protein expressed in the wound epidermis, has been shown to be required for the expression of wound response genes, including *notum* and *wnt1*[75], and a role of *activin-2* in restricting the early expression of *notum* in anterior wounds has been reported[76]. However, the regulation of the expression of *wnt1* in a salt-and-pepper manner in early wounds, and its final restriction to the posterior pole remained unsolved.

Due to the annotation of the CREs in the planarian genome, we identified two putative enhancer regions (E1 and E2) in the first intron of the *wnt1* gene which showed FoxG binding motifs. We propose that these putative enhancer regions are required for *wnt1* expression in planarians. First, they are localized in the first intron, which is a region frequently enriched in regulatory elements[77–80]. Second, we found that *foxG* is necessary for *wnt1* expression in any context. *foxG* inhibition suppresses the early and the late phase of *wnt1* expression during regeneration, in addition to its expression during homeostasis. The different accessibility of E1 and E2 could account for it, since E1 is specifically accessible at 12 hR and E2 is accessible at any time point analyzed. Third, the presence and activity of this enhancer region could be evolutionary conserved, which further supports its relevance. We found that the position of intron 1 in all *wnt1* genes studied in different metazoan species is conserved. Furthermore, there are

several genomic studies that demonstrate the existence of open regions in this intron, and a ChIP-sequencing analysis with the FoxG antibody in *Drosophila* demonstrates that FoxG binds to *Dm-wnt1* (wingless) intron 1. We hypothesize that the binding of FoxG to the intron 1 of *wnt1* to regulate its expression is ancestral and conserved through evolution. Genomic studies in different regenerative species have identified different sets of TFs as regulators of cWnt genes during regeneration. In *Drosophila*, injured imaginal discs required "regenerative enhancers" to trigger *wingless* expression and the regeneration process[81–83]. During *Hydra* head regeneration, an enhancer collection becomes accessible, inducing the expression of cWnt genes in head organizing cells[84–86]. In acoels, *egr* is expressed after amputation triggering the expression of *wnt3*, which participates in posterior specification[82,87]. As recently proposed, it could be that enhancers are maintained as part of conserved gene regulatory network modules during evolution[88]. In this respect, further studies are required to analyze the evolutionary conservation of the enhancers found in the first intron of *wnt1*.

RNAi inhibition of *foxG* suppresses both early and late *wnt1* expression after amputation, and, consequently, animals became tailless. Importantly, a low percentage of *foxG* (RNAi) animals regenerated as two-headed. A shift in polarity is a phenotype only found after *β-catenin-1* or *wnt1* (RNAi)[23,26,30], but never after *islet* or *pitx*

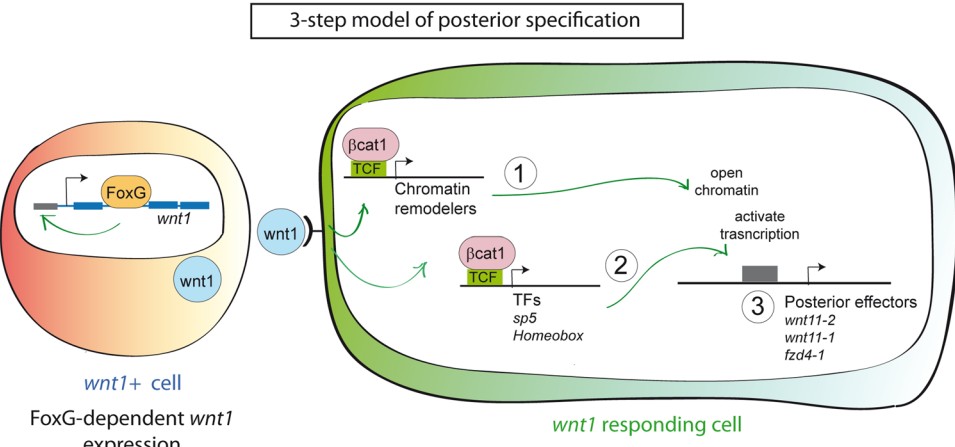

**Fig. 6 | Three-step model of posterior specification.** Our data supports a working model in which after an amputation, *foxG* is expressed at early stages of regeneration in the posterior-facing wounds. The FOXG transcription factor activates *wnt1* expression in the organizer cells, possibly by interacting with an enhancer located in its first intron. Because of this interaction, these cells express and secret *wnt1*. In the 3-step model, the *wnt1* responding cells, through the β-catenin-1/ TCF pathway, will induce the expression of chromatin remodelers (1) and TFs (i.e. *sp5*, *tsh*, Homeobox) (2). Consequently, the chromatin will be accessible for the TFs regulating the expression of crucial genes for posterior specification (i.e. Wnt11s, *fzd4-1*) (3).

(RNAi)[37,38,46]. Thus, the finding of two-headed *foxG* (RNAi) animals suggests that inhibition of the late phase of *wnt1* prevents the regeneration of a tail, but that inhibition of the early phase is required to shift polarity. This idea is supported by reports on the role of Hedgehog (Hh) signaling in planarians. Activation of the Hh signaling is also required for the early phase of *wnt1* expression and a low percentage of animals become two-headed[89,90]. According to these data, Hh could mediate its early role in polarity establishment by regulating *wnt1* expression through *foxG* activation, as has been reported in zebrafish, mice and amphioxus, where Hedgehog signaling contributes to Foxg1 induction and integration of telencephalic signaling centers[91–93].

The planarian posterior organizer is defined by the expression of *wnt1* in differentiated muscular cells. However, the cells in the organizer must integrate a signaling network that includes several genes that are not cell specific, and simultaneously they must be integrated in a diverse and dynamic cellular context. Our study has shed some light on this genetic and cellular context of the planarian posterior organizer. We found *foxG* to be a gene essential for *wnt1* muscle-specific expression, and its inhibition phenocopies *wnt1* (RNAi). Of note, it has recently been shown that *foxG* may play a role in muscle cell specification in many invertebrate organisms, such as planarians[94]. However, *foxG* is not specifically expressed in *wnt1*+ muscular cells but is also expressed in neurons and in progenitor cells. Interestingly, the binding site for FoxG (SLP1) was also notably enriched in the putative enhancer regions of *wnt1* (RNAi) downregulated genes and it is present in the CREs of genes associated with. Thus, FoxG can regulate not only *wnt1* expression in muscular cells but, additionally, *wnt1*-regulated posterior genes in other cell types, such as neurons. Further studies are necessary to analyze whether FoxG binds to E1 and/or E2 of *wnt1*, in addition to the existence of specific co-factors that confer it a cell-dependent activity.

The existence of "regenerative enhancers", groups of enhancers that become accessible during regeneration, has been demonstrated in regenerating species, such as zebrafish and *Drosophila* imaginal discs[81,95–98]. The plasticity of planarians, which allows the comparative study of anterior and posterior regenerating wounds originating from the same field of cells, has allowed the identification of regenerative enhancers specifically associated with posterior specification. The data presented in this study suggests that the formation of the posterior organizer could work as a chain reaction. A first differential signal in the wound according to the polarity of the pre-existing tissue (which could be related to Hh or other neural signals)[89,90] leads to the rapid resolution of the Notum-Wnt1 antagonism, which in posterior wounds will maintain *wnt1* and suppress *notum*. At this point, which must occur during the first 12 h, the program to become posterior has already started, setting up chromatin changes specific to the posterior pole and dependent on cWnt activation. Chromatin accessibility changes allow the subsequent expression of a specific set of TFs that turn on the tail effectors. The finding of several Forkhead binding site motifs in a high percentage of *wnt1* (RNAi) downregulated genes enhancers, as well as in the specific anterior and posterior CREs, aligns with pioneer factors acting at the top of gene regulatory networks to control developmental transitions[99] and prompts to investigate the role of pioneer factors during regeneration and polarity acquisition.

Organizers or organizing centers required for growth and the pattern of a new structure are well studied during embryonic development. Adult organizers are found in whole-body regenerating animals, such as planarians or *Hydra*. Notably, they can be identified both in regenerating and in intact contexts. Muscular *wnt1*+ cells are found in the midline of posterior wounds and in the tail of planarians during homeostasis[24,30], and we found that in both situations *wnt1* expression depends on *foxG*. However, homeostatic and regenerating *wnt1*+ cells have different properties, because inhibition of *wnt1* or *foxG* during homeostasis does not produce a shift in polarity. After an amputation, when new tissue must be regenerated, there must be a time window when everything is possible. According to the signal received by the cells in the wound, the identity of the organizer is decided. Importantly, not only the identity but the presence of an organizer, which indicates the possibility of regenerating, is determined. As shown by Liu et al., modulating the cWnt provided regenerative capacity to planarian species that are not able to regenerate a head in nature[100]. These results indicate that the ability to form an organizer is linked to the ability to regenerate. Thus, understanding the formation and function of organizers is key to understanding adult regeneration, and it must be a subject of study not only in whole-body regenerating animals.

Our results demonstrate the power of genome-wide approaches to further understand the genetics of regeneration. With the aim of sharing the results obtained in this study and facilitating their further analysis by the scientific community, we integrated all transcriptomic and genomic analysis results into the PlanExp open platform (https://compgen.bio.ub.edu/PlanNET/planexp)[43]. Furthermore, an additional

web tool was developed to enable searching for transcription factor binding sites and exploring the predicted regulatory elements (https://compgen.bio.ub.edu/PlanNET/tf_tools).

## Methods

### Planarian husbandry

*S. mediterranea* clonal strain BCN-10 animals were starved for at least 7 days prior to any conducted experiment. Asexual animals were cultured in glass containers and Petri dishes for experiments in planarian artificial medium (PAM) water at 20 °C in the dark. PAM solution contains[101]: 0,016 mM NaCl (Merck, 7647145), 0,01 mM $MgSO_4·7H_2O$ (Merck, 105886), 0,012 mM $NaHCO_3$ (Merck, 106329), 0,001 mM KCl (Merck, 104936), 0,001 mM $MgCl_2·6H_2O$ (Merck, 105833) and 0,01 mM $CaCl_2·2H_2O$ (Merck, 102382) diluted in MiliQ water. Animals were regularly fed twice per week with organic cow liver[102]. Animals with a 5 mm length average were randomly selected for most of the experiments. In the RNA-seq/ATAC-seq and ChIPmentation experiments, animals with 8 mm length average were used.

### RNAi experiment design

For RNAi, double strand RNA (dsRNA) was synthesized by in vitro transcription (Roche) using PCR-generated templates with T7 and SP6 flanking promoters. The precipitation step was conducted using ethanol, followed by annealing and resuspension in water[103]. dsRNA (3 × 32.2 nl) was injected into the digestive system of each animal on 3 consecutive days (1 round). For *wnt1* RNA-seq samples, inhibited and control animals were injected for one round at 1500 ng/µl and amputated at the post-pharyngeal level. Then, studied pieces were soaked in dsRNA diluted (1000 ng/µl) in PAM water for 3 h in the dark. For *wnt1* and *notum* ATAC-seq samples, inhibited and control animals were injected for two rounds at 1000 ng/µl and amputated at pre- and post-pharyngeal levels. *foxG* (RNAi) regenerating animals were inhibited (1500 ng/µl) for two rounds; and high dsRNA concentration (2000 ng/µl) for 5 consecutive days for two weeks was used to increase the RNAi depletion. Animals were amputated pre- and post-pharyngeal at the end of the injection period. Intact animals were inhibited for three consecutive rounds. All control animals were injected and/or soaked with dsRNA of *gfp*.

### Assay for transposase-accessible chromatin sequencing (ATAC-seq)

ATAC-seq samples were obtained from the wound region of wild type, *notum* (RNAi), *wnt1* (RNAi), or *gfp* (RNAi) samples. Twenty animals were used per biological replicate. ATAC-sequencing was carried out as first described in[104] and then adapted by[105]. The ATAC-seq protocol has three main steps: (1) Nuclei preparation. Planarian mucous was removed by washing in 2%L-Cystein (pH 7) for 2 min. Then, animals were transferred to a Petri dish with CMFH (2.56 mM $NaH_2PO_4×2H_2O$, 14.28 mM NaCl, 10.21 mM KCl, 9.42 mM $NaHCO_3$, 1% BSA, 0.5% Glucose, 15 mM HEPES pH 7.3). Planarians were placed in Peltier Cells at 8 °C to amputate the wound region (the blastema and post-blastema region posterior to the mouth). Then, they were transferred to a 1.5 ml Eppendorf tube to be dissociated using a solution of liberase/CMFH (1:10) at RT for 10 min, pipetting until obtaining an homogenized solution. Cell suspension was passed through a 40 µm strainer washed with cold lysis buffer (10 mM TrisHCl pH 7.4, 3 mM $MgCl_2$, 0.2% NP-40, 10 mM NaCl), and cells were quantified using the Neubauer chamber. (2) Transposition reaction. Fifty thousand cells were transferred to a new 1.5 ml tube and resuspended in a transposition reaction (10 µl 2× TD BUffer, 1 µl Tn5 in 9 µl nuclease free water; Nextera XT DNA Library Preparation Kit, FC-131-1024) for 30 min at 37 °C. Then, the samples were transferred to ice. 2.4 µl of EDTA 0.5 M were added per sample, and incubated for 30 min at 55 °C. After that, 1.2 µl of $MgCl_2$ 1 M were added per sample. Following the transposition, the DNA lysate was purified using MinElute Reaction Cleanup Kit (Qiagen, 28206) and

eluted in 10 µl. (3) DNA purification and library preparation. Following the purification, the DNA fragments were amplified using 25 µl of NEBNext High Fidelity 2X PCR Master Mix (BioLabs, M0541S), 10 µl of Transposase DNA, 10 µl of nuclease free water and 2.5 µl of 25 µM adapter 1 and 2.X (Supplementary Data 10); using the following PCR conditions: 72 °C for 5 min; 98 °C for 30 s; 15 cycles of 98 °C for 10 s, 63 °C for 30 s and 72 °C for 1 min; finally hold at 4 °C. The purification was carried out using MinElute Reaction Cleanup Kit, eluting in 10 µl of nucleosome free water. Purification bead step was carried out using AMPure SRI1X to remove small fragments. The libraries were stored at −20 °C. Library preparation and sequencing were carried out by BGI Genomics; libraries were sequenced on HiSeq 4000 (Illumina) with a read length of 2 × 50 bp.

### ChIPmentation

ChIPmentation combines ChIP with library preparation using Tn5 transposase, similar to ATAC-sequencing. ChIPmentation samples were obtained from the wound region of wild type animals. A total of 2000 anterior and posterior blastemas were used. Groups of 100 blastemas were done at one time. The ChIPmentation was carried out as described in ref. [106] and it has four main steps: (1) Tissue crosslinking. Planarians were placed in Peltier Cells at 8 °C to amputate the wound region (the blastema and post-blastema region posterior to the mouth). Then, wounds were transferred to a Petri dish containing 1 M $MgCl_2$ solution, for 15–30 rocking at room temperature (RT). PBS 1× was added to remove salts. Blastemas were fixed with formaldehyde 1.85% for 15' rocking, at RT. Glycine was added to obtain a final concentration of 0.125 M to quench formaldehyde, for 5' at RT, rocking. Then, blastemas were washed 3× with cold PBS 1×. Finally, PBS excess was removed, and samples were stored at −80 °C. (2) Tissue homogenization, sonication and immunoprecipitation. Samples were homogenized in a lysis buffer (10 mM TrisHCl pH 7.5, 10 mM NaCl, 0.3% NP-40, Complete 1×). After centrifuging the samples, the pellet was resuspended in 660 µl Nuclear lysis buffer (50 mM TrisHCl pH 7.5, 10 mM EDTA, 1 % SDS, Complete 1×) and 1.34 ml of ChIP Dilution buffer (16.7 mM TrisHCl pH 7.5, 1.2 mM EDTA, 167 mM NaCl, 0.01 % SDS, 1.1% Triton X-100). Then, samples were sonicated in a M220 Focused-ultrasonicator (Covaris, 500295) with the following settings: duty factor = 10%, PIP = 75 W, cycles = 100, time = 10 min. The sonicated chromatin was incubated with H3K27ac polyclonal antibody (1:100; abcam, ab4729) overnight at 4 °C. (3) Bead incubation and TAGmentation. The samples were incubated with Dynabeads protein G resuspended in ChIP Dilution Buffer for 1 h at 4 °C. Then, beads were washed twice with Wash Buffer 1 (20 mM TrisHCl pH 7.5, 2 mM EDTA, 150 mM NaCl, 1% SDS, 1% Triton X-100), twice in Wash Buffer 2 (20 mM TrisHCl pH 7.5, 2 mM EDTA, 500 mM NaCl, 0.1% SDS, 1% Triton X-100), twice in Wash Buffer 3 (10 mM TrisHCl pH 7.5, 1 mM EDTA, 250 mM LiCl, 1% NP-40, 1% Na-deoxycholate), and twice in 10 mM TrisHCl pH 8. Beads were resuspended in TAGmentation reaction mix (10 mM TrisHCl pH 8.0, 5 mM MgCl2, 10% w/v dimethylformamide) and 1 µl of Tn5 and incubated for 1 min. Subsequently, two washes with Wash Buffer 1 were performed and once with TE 1×. The samples were eluted 15 min with 100 µl of Elution Buffer (50 mM NaHCO3 pH 8.8, 1% SDS) and finally with 180 µl of Elution Buffer Per sample, 10 µl of 4 M NaCl and 0.5 µl of 10 mg/ml ProteinaseK were added, being incubated between 4/6 h at 65 °C. (4) DNA purification and library preparation. DNA was purified using Minelute columns (Qiagen, 28004). The libraries ware prepared by PCR reaction: 19 µl of DNA, 1 µl of 25 µM of adapter 1 and 2X (Supplementary Data 10), 25 µl of 2X NEBNext High Fidelity Master Mix and 4 µl of MQ water; using the following PCR conditions: 98 °C for 30 s; X cycles of 98 °C for 10 s, 63 °C for 30 s, 72 °C for 30 s; and 72 °C for 5 min; finally hold at 4 °C. The number of cycles for library preparation was empirically determined by qPCR. Finally, the libraries were purified using Minelute columns. From each 1/100 ChIP library, 1/10, 1/100 and 1/1000 dilutions of the inputs were used in a qPCR. To test the

generated libraries, positive and negative primers were designed (Supplementary Data 10). Library preparation and sequencing were carried out by BGI Genomics; libraries were sequenced on HiSeq 4000 (Illumina) with a read length of 2 × 50 bp.

## ATAC-seq and ChIPmentation analysis

Reads were aligned by bowtie[107] using -m 3 -k 1 arguments. Reads were filtered from BAM files using a ≤ 100 bp insert size threshold to identify nucleosome free regions (NFRs)[108]. BAM were converted to BED; for the ATAC-seq replicates then the coordinates on the BED files were shifted +4 and −5 positions to overcome the Tn5 cut position. MACS2[109] was used for peak calling and HOMER[110] for motif discovery. Differential binding analysis was carried out using DiffBind[111] R functions (see CSM sections 4 and 5).

## ATAC-seq comparison ACR between anterior and posterior

Significant differentially bound sites obtained by MACS2 from ATAC-seq anterior versus posterior comparison, 611 and 2484 respectively, were crossmatched by DiffBind against the corresponding significant MACS2 differentially bound sites from anterior and posterior ChIP-seq control samples. Significant sites computed over those site sets by DiffBind (FDR < 0.05), were assumed to be promoter elements; on the other hand, such procedure returned 555 and 1869 non-significant sites that were marked as putative enhancers for anterior and posterior respectively. Those enhancer regions will be referred to as ATAC-ChIP peaks. Enhancer dynamics was derived after crossing with DiffBind the data of anterior and posterior ATAC-ChIP peaks over *notum* and *wnt1* (RNAi) ATAC-seq samples (DiffBind "second round" from now on). All anterior and posterior ATAC-ChIP peaks found non-significant on the "second round" that had positive counts when compared with the two replicates of the *notum* or *wnt1* (RNAi) ATAC-Seq peaks were considered "accessible". If an ATAC-ChIP peak was significant on the "second round" and both replicates of ATAC-ChIP and both of *notum* (RNAi) ATAC-Seq had positive counts, then those peaks were annotated as "slightly" for posterior regenerating blastemas or "less" accessible for anterior regenerating blastemas. If an ATAC-ChIP peak was significant on the "second round" and both replicates of ATAC-ChIP and *wnt* (RNAi) ATAC-Seq samples had positive counts, then those peaks were annotated as "slightly" for anterior regenerating blastemas or "less" accessible for posterior regenerating blastemas. Finally, all those peaks found significant on the "second round" that had two positive ATAC-ChIP replicates and less than two of *notum* or *wnt1* (RNAi) ATAC-Seq sample replicates, were annotated as "non-accessible" (see Supplementary Data 1 and further details on CSM, section 5.4 and Figure CSM.29).

## Cis-regulatory elements annotations

Putative *cis*-regulatory elements (CRE) were annotated over the *S. mediterranea* genome version S2F2. For this purpose, both ChIP-seq and ATAC-seq data from all collected samples were used. Narrowpeaks over the genome were identified using MACS2 (see "ATAC-seq and ChIPmentation analysis" section of "Methods"). These peaks were merged using the mergePeaks command of the HOMER software suite. Finally, regions over the genome were classified as either putative promoters, or putative enhancers, according to their evidence regarding ATAC-seq and ChIP-seq peak coverage. Only the regions known as peaks on at least two samples were considered, and the remainder were discarded for the CRE annotation (see CSM, section 7).

Peak regions with only ATAC-seq evidence were classified as core promoters (<100 bp upstream of an annotated TSS) or proximal promoters (between 500 and 100 bp upstream of a TSS). Finally, peaks with CHIP-seq evidence were classified as either proximal enhancers (within 2000 bp of an annotated TSS) or distal enhancers (between 2000 and 10,000 bp of a TSS).

## Motif finding

Putative transcription factor binding sites were identified and annotated on all of these enhancer and promoter regions using the HOMER's findMotifsGenome command, scanning these regions using the motifs provided by the software suite (known motifs). Two further motifs were manually appended to the set of known motifs: slp1 from *Drosophila melanogaster* (MA0458.1) and FOXG from *Homo sapi*ens (MA0613.1). Weight matrices were obtained from the JASPAR CORE 2022 database[112].

HOMER findMotifsGenome command computed motifs overrepresentation within those enhancer and promoter regions with respect to random sampled regions scattered through genome. All motifs considered from that output had an adjusted p-value smaller than 0.05. Then, enhancer and promoter regions were assigned to the closest gene, which facilitated the calculation of ratios of FoxG motif matches with respect to the total number of motifs found on each region for each gene (see Figure CSM.47). At 12 h anterior/posterior comparison 4035 regions were assigned to genes; of those, 3290 had no FoxG motif annotated in either promoter or enhancer regions (top panel on Figure CSM.49), 304 were found to have FoxG motif only on anterior samples ("A to 0"), whilst 433 only on posterior samples ("0 to P"), and the remaining 8 genes had similar anterior/posterior FoxG ratios. See CSM sections 7.4.5 for further details on the gene selection related to the anterior/posterior FoxG enrichment ratios calculation, and 7.5.3 for the gene ontology functional analysis over those genes (highlighted on Supplementary Data 9 and Supplementary Fig. 10).

## ACR/Enhancer/FoxG binding site motif research in different species

The presence/absence of accessible chromatin regions and enhancer regions in different species was determined by visualizing the ATAC-seq and DNase-seq, and H3K27ac and H3K4me1/2/3 tracks, respectively. Genome Browsers used were listed in Supplementary Data 7.

The presence of the FoxG binding site motif was determined using the FIMO tool (https://meme-suite.org/meme/tools/fimo)[113] using the default parameters. FASTA files containing DNA sequences corresponding to the *wnt1* first intron from different species were obtained from different sources, mostly using UCSC browser (Supplementary Data 7).

## Integration with online resources

A new plugin for the PlanNET web service[114], called "TF Tools", was developed to integrate the putative CRE dataset with existing planarian resources. A search tool for exploring genes according to the presence or absence of transcription factor binding motifs was developed, and the putative CRE elements were incorporated into the existing gene cards in PlanNET and to our genome browser instance. The website, the source code of the plugin, and downloads for all the annotations are available at https://compgen.bio.ub.edu/PlanNET.

## RNA-sequencing sample preparation and analysis

RNA-sequencing samples were obtained following the soaking protocol. At the corresponding time point (0, 24, 48, and 72 h of regeneration), animals were placed in a Petri dish with cold 1% HCl (diluted in water) for 2′ and then transferred to a new Petri dish with cold PBS 1×. Two washes were performed with cold PBS 1× and animals were transferred to cold RNAlater for 20′ placed in ice. Then, planarians were amputated in a Peltier Cell with a clean blade, to obtain the blastemas and post-blastemas. Fragments were washed with RNAlater (Invitrogen) and 50% RNAlater/Trizol. Finally, liquids were removed and 100 μl of Trizol reagent (Invitrogen) was added. Total mRNA extraction was performed as described in ref. [115]. Three biological replicates were used per time point. Each biological replicate was composed of eight animal fragments. Library preparation and sequencing were carried out by the Centre Nacional d'Anàlisi Genòmic

(CNAG); libraries were sequenced on HiSeq 4000 (Illumina) with a read length of 2 × 76 bp.

RNA reads were mapped against the planarian genome version S2F2[42] using the STAR software tool[116]. Genes with low expression were filtered by removing genes with less than 1 count-per-million (CPM). Two biological replicates were removed due to ineffective *wnt1* inhibition (see CSM, section 3). Differentially expressed genes were identified using the lima-voom pipeline[117], using an FDR cut-off of 0.05 and a log fold change cut-off of ±0.5. Interactive expression analyses are also available from PlanEXP as "2019 Adell Time-course" dataset.

The day 1 (42), day 2 (95) and cluster (52) of downregulated genes from RNA-seq of *β-catenin-1* (RNAi)[29] were used to combine with the downregulated genes *wnt1* (RNAi) wounds. The transcriptome IDs were converted to genome IDs, using the "ID Converter" tool from PlanNET, and used to determine their presence in our dataset.

### Whole mount in situ hybridization (WISH)

RNA probes were synthesized in vitro (Roche) using T7 or SP6 polymerases and DIG-modified, purified with ethanol and 7.5 M of ammonium acetate, diluted in 25 μl ddH$_2$O and adjusted to a final concentration of 250 ng/μl. All primers used are displayed in Supplementary Data 10. For colorimetric ISH, the following was performed[53]: animals were killed in 5% N-acetyl-L-cysteine (NAC), fixed in 4% formaldehyde (FA), permeabilized with reduction solution for 5' at 37 °C and stored in methanol at −20 °C. Following overnight hybridization, samples were washed twice with 2x SSC with Triton-X (SSCTx), 0.2x SSCTx, and 0.02x SSCTx and MABTween. Subsequently, blocking was in 5% Horse Serum, and 0.5% Western blocking reagent (Roche) MABTween solution and anti-DIG-AP (1:4000, Roche; 11093274910) was used. The antibody was washed for 2 h followed by NBT/BCIP development.

### Immunohistochemistry staining

Whole-mount immunohistochemistry (IF) was performed as in ref. [118]: animals were killed with cold 2% HCl and fixed with 4% FA at RT. After 4 h in blocking solution (1% BSA in PBS Triton-X 0.3%), animals were stained overnight at 4 °C. Animals were washed extensively with PBSTx, blocked for 2 h, and stained overnight at 4 °C. The following antibodies were used in these experiments: mouse anti-synapsin (anti-SYNORF1/3C11, 1:50; Developmental Studies Hybridoma Bank) and anti-Smed-β-catenin-2 (1:1000;[119]). The secondary antibodies used were Alexa 488-conjugated goat anti-mouse (1:400; Molecular Probes; A28175) and Alexa 568-conjugated goat anti-rabbit (1:1000; Molecular Probes; A-11011). Nuclei were stained with DAPI (1:5000).

### Quantitative real-time PCR

Total mRNA was extracted from a pool of 5 control and RNAi planarians at 7 days of regeneration, using TRIzol reagent (Invitrogen). RNA samples were treated with DNase I (Roche, 4716728001) and cDNA was synthesized using First-Strand Synthesis System Kit (Invitrogen, A48571). Experiments were conducted on 7500 Fast PCR System (Applied Biosystems), using 3 biological and 3 technical replicates for each condition. The housekeeping *ura4* was used to normalize the expression levels, all primers used are displayed in Supplementary Data 10.

### Image acquisition

In vivo images were acquired with Scmex 3.0 camera (DC.3000s, Visual Inspection Technology) in a Zeiss Stemi SV 6 binocular loupe. Brightfield colorimetric ISH images obtained with a ProgRes C3 camera from Jenoptik (Jena, TH, Germany). A Zeiss LSM 880 confocal microscope (Zeiss, Oberkochen, Germany) was used to obtain confocal images of whole-mount immunostainings. Fiji/ImageJ[120] was used to show representative confocal stacks for each experimental condition.

### Statistics and reproducibility

Information regarding statistical tests, n values, and replicates are detailed in figure legends, Computational Supplementary Methods (CSM), Source Data and Reporting Summary. Two-sided Student's t-tests ($α = 0.05$) were performed to compare the means of two populations (qRT-PCR experiment). P values of <0.05 (*), <0.01 (**), <0.001 (***) were annotated with asterisks in figures. Statistical analyses were performed using GraphPad Prism 8 (GraphPad Software, San Diego, CA). For in vivo, ISH and IF experiments, values indicate exactly the animals used in each experiment. The experiment reproducibility is indicated either in the figure legend or the CSM.

Four RNA-Seq replicates were discarded: two sequencing libraries due to low sequencing yield (control 12 hR and 36 hR, see Figure CSM.5), which were compensated by an extra replicate at those timepoints; and two further replicates later on (*wnt1*-RNAi 0 hR and 12 hR, see Figure CSM.10), because their *wnt1* expression level was overlapping the one of the control, which indicates ineffective *wnt1* inhibition on those samples. Therefore, all RNA-Seq timepoints consisted of three replicate libraries for both controls and *wnt1* (RNAi) samples, except for *wnt1* (RNAi) 0 hR and 12 hR that had only two. ATAC-Seq single replicate timepoints at control 0 hR, anterior 48 hR, and posterior 48 hR, were not considered to analyze the dynamics of regulatory elements. Three library replicates per ATAC-Seq condition at 12 hR were obtained; however, one replicate of each condition was discarded after PCA analysis (as described in Figure CSM.21). All the downstream analyses were made using two replicates per condition of the ATAC-Seq at 12 hR and for the ChIPmentation anterior/posterior conditions; on all those replicates FRiP score was greater than 0.3 (ATAC 12 hR replicates > 0.82, ChIP replicates > 0.85; see for instance Figure CSM.20), and irreproducible discovery rate (IDR) rank plots showed good correlation between peaks annotated on replicate pairs (see Figures CSM.82 and CSM.89; and tables CSM.T8 and CSM.T11). Replicate and statistical information for RNA-seq, ATAC-seq, Chipmentation data and other bioinformatic analysis are detailed in the CSM.

### Reporting summary

Further information on research design is available in the Nature Portfolio Reporting Summary linked to this article.

## Data availability

The raw sequencing data sets generated for this research have been deposited with links to BioProject accession number PRJNA800775 in the NCBI BioProject database. The corresponding NCBI Sequence Read Archive (SRA) accessions for the samples included in that BioProject are: for RNA-Seq samples, SRR17766314, SRR17766312, SRR17766311, SRR17766310, SRR17766301, SRR17766300, SRR17766299, SRR17766298, SRR17766297, SRR17766296, SRR17766295, SRR17766325, SRR17766324, SRR17766323, SRR17766322, SRR17766321, SRR17766320, SRR17766319, SRR17766318, SRR17766309, SRR17766308, SRR17766306, SRR17766305, SRR17766304, SRR17766303, SRR17766302, SRR17766286, SRR17766285, SRR17766284, SRR17766283, SRR17766282, SRR17766280, SRR17766279, SRR17766293, and SRR17766292, SRR17766291; for ATAC-Seq samples, SRR17766333, SRR17766328, SRR17766327, SRR17766332, SRR17766313, SRR17766294, SRR17766289, SRR17766288, SRR17766287, SRR17766307, SRR17766281, SRR17766290, SRR17766331, SRR17766330, and SRR17766329; finally, for ChIP-Seq samples, SRR17766326, SRR17766317, SRR17766315, and SRR17766316. Other data supporting this

study's findings are available within the article and its Supplementary files. Source data are provided with this paper.

## Code availability

A PDF describing all the Computational Supplementary Methods (CSM) is available from GitHub together with the scripts used for the Bioinformatic analyses (https://github.com/CompGenLabUB/2022_NatComm_BImethods, https://doi.org/10.5281/zenodo.7455255).

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

## Acknowledgements

We wish to thank all members of the Emili Saló, Teresa Adell, and Francesc Cebrià labs for their suggestions and discussion of the results. We thank Susanna Balcells and Núria Marínez Gil for their help in setting up the analysis of the Wnt1 enhancers. A thought to the memory of José Luis Gomez Skarmeta, who passed away on September 16, 2020, while this manuscript was in progress; he was an outstanding researcher and an exceptional promoter of scientific collaborations. E.P.-C. is a recipient of an FPI (Formación del Profesorado Investigador) scholarship from the Spanish Ministerio de Ciencia, Innovación y Universidades. S.C.-L. was a recipient of a FI-FDR fellowship 2017FI_B_00191 from AGAUR (Generalitat de Catalunya). M.M.-B was supported by the People Programme (Marie Curie Actions) of the European Union's Seventh Framework Programme FP7 under REA grant agreement number 607142 (DevCom). M.M.-B also thanks the John and Pamela Salter Trust for funding support. E.S. and T.A. received funding from the Ministerio de Educación y Ciencia (grant number BFU2017-83755-P, BFU2014-56055-P and BFU2020-116372GB-I00). E.S. and T.A. benefit from 2017SGR-1455 from AGAUR (Generalitat de Catalunya). E.S. received funding from AGAUR (Generalitat de Catalunya: grant number 2014SGR687). J.L.G.-S. received funding from the ERC (Grant 944 Agreement No. 740041), the Spanish Ministerio de Economía y Competitividad (Grant No. 945 BFU2016-74961-P) and the institutional grant Unidad de Excelencia María de Maeztu (MDM946 2016-0687).

## Author contributions

E.P.-C. and T.A. designed the study and wrote the manuscript; E.P.-C., M.M.-B. and S.C.-L. performed and analyzed genomic experiments; P.C.-C performed *foxG* (RNAi) experiments; J.F.A., S.C.-L. and M.M.-B. performed the bioinformatic research; J.F.A supervised the bioinformatic research. M.M.-B., M.S.M., G.N.W. and J.L.G.-S. performed the ChIP-seq experiments and analyzed the results; E.S. and T.A. received the funding and supervised the research.

## Competing interests

The authors declare no competing interests.
