## [Peer Review File · Nature Communications]

Wnt/ β -catenin signalling is required for pole-specific chromatin remodeling during planarian regenerationREVIEWER COMMENTS

Reviewer #1 (Remarks to the Author):

The submission from Pascual-Carreras et al integrates genome wide analyses that attempts to identify cis-regulatory elements during early regeneration in planarians, interpret this data in light of RNAi of key genes, and correlate this with later gene expression changes. They also provide the very interesting FoxG(RNAi) phenotype and attempt to link this to the regulation of wnt1, this data is relatively compelling. This a potentially very exciting and pioneering study for this model organism. Overall, in its current form the work is difficult to follow, there is little in terms of data quality control presented (although this may have been performed), and as a result the analysis of data requires some further work and hypothesis testing. There is no assessment of the the quality of the ATAC/ChiP data or the statistical significance of their results/analyses relative to any null hypothesis. While they use standard approaches and software there needs to be a more careful assessment of the significance of the data and more detail and clarity presented on how data was analysed. This issue needs to be addressed before this work can used as the basis of future work (and hence peer reviewed publication). They need to perform and present these QC analyses to demonstrate their findings, regarding CRE identification, motif enrichment and subsequent inferred regulatory interactions are potentially real. I should note that it seems likely they could be as there much of what they have found would be predicted by existing work in the literature. In order to help I have tried to outline what some of these QCs and analyses would below. I also have a number of comments, suggestions and questions about the data which I hope will be helpful.

Comments about experiments/data in the paper as presented.

1. Why did the authors only look at RNAs-seq at 0, 24, 48 and 72 hours of regeneration after wnt1 RNAi. Why not use matched 12 hour samples for comparing to ATAC-seq experiments? It seems that this would be the most informative timepoint.
2. Do the authors have ATAC-seq data for un-wounded tissue as a good control data set? This would allow them to set strict criteria for changes in accessibility.
3. The FoxG data with respect to establishing direct regulation and the potential broader role of FoxG is incomplete if the reader is to assess the evidence properly. What level is the H327ac signal at? Is this significant? Is ATAC data the same between both wounds or different? Are the peaks containing the FoxG motif called as significant by the methods presented in the paper? What other motifs are there in these enhancers? Given the significance of this for the authors manuscript/narrative I would suggest presenting the raw mapped read data for both signals, and providing a measure of their significance “how likely are they to be active enhancers
4. Do ACRs/Enhancers identified in the first intron of Wnt1 of other species also contain FoxG binding motifs? This is not clear in the paper as presented, this should be shown. Does the evidence for ACR and enhancer in the first intron overlap (along with a potential binding site?)

5. What other potential FoxG targets are implicated by this simple method, and are they also wound induced genes?

6. The FoxG expression pattern seems to disperse as presented, is there expression in cell at the anterior margin?

7. Would the authors be able to look at ATAC signal in FoxG(RNAi) animals to see if the signal changes? Generally, Fox family TFs are classed as pioneers.

Comments on the quality control of data and data analysis approaches.

ATAC-seq Quality control.

1. Given this is one of the first, and the first extensive examples of ATAC-seq on planarians the authors need to supply more quality metrics for ATAC-seq, including tagmented library profiles and measures of the proportion of mapped reads under peaks. This will allow readers to assess the quality of libraries, and the level of tagmentation in these libraries. They also need to show the global level of peaks around their center of highest coverage and describe the distribution of peak widths, showing which are significant (called as peaks). They should also present the ATAC accessibility around predicted TSSs across the genome. This will establish the ATAC data is reasonable, not over tagmented (although this can be useful too).

2. The authors should present biological replicate and experimental condition correlation plots to look at the global congruence between replicate peak calling, to get more assurance that the approach is working well.

3. Also they should include a comparison to previously published ATAC data as a QC step (Gehrke et al, 2019), some of the accessible regions should be the same.

4. The authors present data of ATAC peaks that show changes in accessibility after RNAi. These numbers suggest changes in most of the identified posterior or anterior specific CREs. I am not clear on how these changes in accessibility were measured or assessed? What were the criteria for these classes? The authors need to explain this and present some specific examples in the paper. They should also clarify the overlap in these changes with Venn diagrams as we might expect them to change in opposite directions for some peaks?

5. The authors refer to previous ATC and and Chipmentation samples (0-48 hours) what are these precisely? Is there a reference for them?

Chipmentation Quality Control

Similar to the ATAC-data, the authors need to show controls and QCs for this approach. There are a number of possibilities.

1. They could include comparison to other available data sets, even if the H3K27ac is not available there are other marks that they could look at for correlation.
2. The correlation between these data and ATAC data should be considered globally, how many peaks do not overlap, is the overlap greater than that expected by chance?
3. Are ATAC peaks bounded by H3K27ac peaks? (the authors are aware of this pattern as they use it later in the paper).
4. Is their correlation with, for example, H3K4me1 signal? These data sets do exist in some form for planarians and are available in the short-read archive.

Calling CREs and potential Transcription factors at 12 hours, and then at later timepoints.

The authors need to query their data for noise to signal. For example, looking for motifs in random blocks of sequence amongst the same genomic regions around differentially expressed genes vs all genes, or just in ATAC peaks or H3K27ac peaks. The second set of CREs discovered using ATAC at different timepoints is not supported by chipmentation data as far as I can ascertain, so again it is very important these data are backed up by rigorous QC and some analysis to show that motif enrichment under peaks is distinct from motif enrichment in the same region with low/zero ATAC coverage. These later ATAC data also need standard QC as described above for the 12-hour timepoints.

To illustrate this can the authors show the presence of potential TCF sites around potential target genes to see if they are actually concentrated under peaks, look at peaks in genes that are expressed but not differentially and show they don't have TCF site enrichment, similarly for genes that are up-regulated what is the collective presence or absence of TCF sites? Without these analyses the authors only look for what they wish to see and have no idea if it is meaningful or just chance occurrence. For the homeobox analysis TCF sites also need to be investigated in the homeoboxes that do not appear to be regulated with respect to genes expression analysis- these should not have TCF sites.

The motif enrichment analysis should also be presented for up-regulated genes as a comparison.

To make the data useful somewhere all the planarians orthologs to the transcription factors presented should be presented for the reader (where they actually exist, i.e. not Nanog!)

Minor comments on the clarity.

The paper could do with significant editing for clarity and grammar to help the reader.

Abstract

The abstract could be re-written to more clearly reflect the results. To my knowledge "wound cells" is not a working definition of any cell type or state. Do the authors mean that "amongst cells at the wound site chromatin accessibility and histone marks change between anterior and posterior facing wounds".

This also appears in the introduction line 84, and could be clearer for non experts.

The authors study chromatin conformation by the normal definition as direct contacts between enhancers/promoters etc are not measured using a Hi-C type approach. Through the paper conformation should probably be changed to accessibility or ATAC-signal.

"pre-existent" should be "pre-existing"

"WNT/beta-catenin signal" should be "signalling"

I would advise that the final point regarding FoxG is worked into the main body of the abstract as this

the major implication of this of this work rather than just a “Furthermore,”

Similar clarity issues exist through the manuscript that could be improved to help the reader before any subsequent submission.

Introduction

Probably pertinent to mention the recent description of activin-2 as being required for the asymmetric expression of Notum (ref) (line 72) i.e. something is now known about this.

Results

See above, but clarity of message could be improved for the reader with careful editing.

The discussion is well written and comprehensive, if rather selective in mentioning transcription factors that control anterior and posterior regeneration.

--

Reviewer #2 (Remarks to the Author):

In this manuscript, the authors explore polarity reestablishment during planarian regeneration with a focus on how chromatin reorganization might play a key role. The first half of this paper is a tour de force of methods that include ATAC-Seq and ChIPmentation, which are new or nearly new to the planarian community. The authors also use RNAi/RNA-Seq and site prediction strategies to connect their chromatin discoveries to Wnt/Beta-catenin signaling and the cellular/molecular basis of reestablishment of anteroposterior polarity. In the second half of the paper, the group discovers a transcription factor, FoxG, that is important for expression of Wnt-1, a key signaling molecule in the A/P signaling process. The mechanism for this regulation appears to be well-conserved. There are many aspects of this work that are novel and exciting and the use of new techniques is innovative, as well. However, the two halves of the paper are not clearly linked. Further, the work requires several major changes or additions before it would be appropriate for publication in Nature Communications.

Major issues:

1) The most difficult flaw in this paper is that the two halves of the paper are not well connected, either in writing or conceptually. The first half of the paper seeks to explore changes in chromatin status that occur as a result of Wnt/Beta-catenin signaling perturbation, but then the second half of the paper swings around to explore how Wnt1 is itself controlled by FoxG, a series of results that are interesting but fully unconnected to the chromatin reorganization story. One solution to this problem would be to separate this paper into two; this would depend on the wishes of the authors and editors. Alternatively, the authors would need to rework the manuscript or add data so that the manuscript works well together.

2) The title indicates that “Wnt/beta-catenin signal controls pole-specific chromatin remodeling during planarian regeneration.” The “control” element of this claim is not well supported. The authors do demonstrate a correlation between chromatin changes and Wnt/Notum perturbation, but the mechanism connecting Wnt and chromatin is not at all explored.

a. The authors claim that some chromatin remodeling factors are downregulated after Wnt(RNAi), but the fold changes for the two genes mentioned are modest and no functional data are included to show that BPTF and TOP2B have any role in chromatin organization changes seen after Wnt perturbation and/or polarity of the planarian body. Identification of the Wnt-regulated factor that causes A/P chromatin changes would be ideal.

b. Though chromatin remodeling or modifying enzymes might be targets of Wnt signaling, another possibility is transcription factors downstream of Wnt signaling (either beta-catenin/TCF? or TFs differentially expressed downstream of Wnt) act as pioneer TFs to recruit chromatin modifying/remodeling enzymes to particular regions of the genome. This possibility could also be explored bioinformatically and experimentally.

3) The methods are not well-described, in terms of technique or data analysis. This needs to be improved for publication, particularly since these are newer methods for planarians and some of this paper’s impact comes from the establishment/adaptation of new methods. The citations provided are not for planarian samples. Information about controls or evaluation of quality of data are not included. In addition to a general improvement in the methods section, the following are important both for reproducibility and interpretation of data.

a. Because planarian promoters are not at all well characterized, more discussion should be included about how peak calling was made, how peaks were assigned to nearby genes (for example when peaks are near more than one gene), what controls were used, how proximal/distal promoter definitions were defined, etc. A general discussion about trends seen for planarian promoters (in the text or in the supplement) might also be warranted, as this information could be more broadly valuable to others trying to understand gene regulation and promoter structure in planarians.

b. ChIPmentation methods should be extensively detailed, since this is the first use of this method in planarians (to my knowledge). The H3K27ac antibody used should be stated. Controls for this particular experiment should be explained.

c. The authors should clarify to the reader what criteria were met for a region to be called an “enhancer,” as this term has a particular functional connotation and function of these sequences have not yet been studied. The authors could consider using a less functional term “accessible region” until function is determined.

d. Further, how distinctions for enhancers like “accessible/slightly accessible/less accessible/non-accessible” were made warrant clearer definition. What are thresholds for these different categories?

e. The ATAC-seq analysis of 0 and 48 h wounds are referenced, but only briefly. This may also warrant further explanation. It isn’t clear why these analyses were included and how these results were integrated with other results.

f. The authors should clarify to the reader which approaches are novel and which have been used for planarians before. This will help the reader appreciate the novelty of this work, which might be overlooked otherwise. Where other similar approaches have been used, adding more clarity will also give appropriate credit to other related work and help put this manuscript in context.

4) It bears clarifying that the transcription factor binding site enrichment approach is based on the binding sites of TFs from other organisms. The TF binding sites for planarian TFs have not been determined. Therefore, some results from this approach may well indicate conserved binding sites, but others could be misleading. Care should be taken to avoid overinterpretation based on these binding sites without experimental validation.

a. One way to connect the two stories would be to actually characterize the accessible elements and binding sites for wnt 1, experimentally verify the FoxG binding site, and assay whether FoxG does bind to the wnt1 promoter and elsewhere.

5) A few questions about FoxG remain:

a. foxG expression in Fig. 3d appears to be fairly ubiquitous. Can the authors explain how FoxG promotes Wnt1 expression in such a specific location despite FoxG being present elsewhere? Where/when is foxG expressed in regeneration? Is it also coexpressed in anterior wnt1+ cells?

b. Are the predicted FoxG binding sites (in wnt1 or elsewhere) differentially accessible in posterior samples from the ATAC-seq data?

c. The foxG(RNAi) experiments in Fig. 5b (especially hox4b and sp5) are not always clear would be more quantifiable (with statistical analysis) with RT-qPCR.

Minor issues:

1) N for several experiments is needed (in figure legends, for example in 5A, 5C). N should be added throughout.

2) ATAC-Seq or similar methods performed on complex tissues probably misses some cell type-specific complexity in chromatin arrangement. It could be useful to discuss limitations of these approaches in the discussion. For example, are “anterior” or “posterior” specific peaks expected to be the same in all cells or changing only in specific cell types (e.g. muscle)?

3) In Fig. 1A, it isn't clear how many peaks were in both A & P samples.

4) Figure 4 is fairly confusing and the binding sites and conservation of exons/binding sites could be more clearly demonstrated. I think my confusion arises because the intron is removed in A but is the focus of the conservation discussion.

5) The authors state in the introduction that the existence of organizing centers in adults “has not been well studied,” but this is fairly well studied in planarians. Consider rephrasing?

6) The plural of motifs is usually with an F.

--

Reviewer #3 (Remarks to the Author):

The manuscript entitled “Wnt/ β -catenin signal controls pole-specific chromatin remodeling during planarian regeneration” by Pascual-Carreras et al., is a timely and exciting contribution to the field of regeneration and chromatin biology. Although not directly addressed, the manuscript also raises intriguing questions about the role of ‘pioneer’ transcription factors in establishing polarity during regeneration.

The Key Results of the manuscript are (1) that wnt and notum, which are known to be required for establishing polarity during regeneration in planarians, are required for chromatin remodeling and enhancer accessibility early after injury (<12 hours) well before the 1st anterior and posterior markers appear; (2) that the transcription factor FoxG is required for wnt expression in posterior and anterior wounds and for proper polarity after injury; and (3) that intron 1 of wnt bearing chromatin accessible regions with enhancer activity is evolutionarily conserved.

Broadly speaking, the approach and experimental design are sound; however I have a few major and minor points regarding additional analyses, data clarity and reproducibility, and suggestions for providing appropriate context in the text, that I believe would substantially improve the manuscript and should be addressed before acceptance. It is my belief that these are not extraordinary requests and that the authors can complete them in a timely manner so that the community can appreciate this important contribution.

Major points:

1. The manuscript would significantly benefit from a t'0 enhancer accessibility time point (prior to or immediately after injury). Are the enhancers identified in this manuscript specific to establishing polarity in regeneration, or do they just provide homeostatic A and P ends? It seems that such data might already exist and be publicly available? I highlight a sentence from their summary of chromatin accessibility (and the title of Figure 1) to emphasize this point:

“the chromatin structure of cells in the wound is remodeled according to the polarity of the pre-existing tissue, which occurs a few hours after amputation (<12 hR), before the first anterior or posterior markers appear (around 48 hR).”

Is it really remodeled in response to injury, or was it already there?

*Although it may not be this simple, the authors might also find that the FoxG enhancer in intron 1 of Wnt is closed prior to injury, and that the opening allows FoxG to bind, induce Wnt expression, and initiate posterior regeneration, which I believe would add an additional layer to their manuscript.

2. There is not sufficient description of the bioinformatic methodology for assigning the different levels of chromatin accessibility in Figure 1. What distinguishes 'Accessible' from 'Slightly' and 'Less' accessible (also there is a typo in the figure legend that reads 'Slightly' accessible). As this is a critical component of the manuscript, this needs to be addressed.

1b. I know it's simple, but I think the manuscript would benefit from a brief description of the pole-specific filter to establish A- and P-specific enhancers in the methods section.

3. Related to point #1; there seems to be a discrepancy between the reported enhancer accessibility and the Figure 1B panel. First, again, it is unclear what slightly vs. less accessible means. Second, the authors

state that:

“in notum (RNAi) anterior wounds, only 12.3% of the anterior putative active enhancers were open, whereas the remainder were closed or reduced their accessibility”,

however the pie chart in Figure 1B shows >25% accessible for this sample. Either the pie chart is only representative, in which case it should be re-made to reflect the data, or there is a discrepancy between the data and figure panel.

Third, the authors state:

“Furthermore, 31.4% of the anterior putative active enhancers appeared to be open in wnt1 (RNAi) posterior wounds and 9.5% became more accessible”,

however the pie chart shows maybe 31.4% as “less accessible” rather than “accessible”, and maybe 9.5% as “accessible” rather than “more (?) accessible”.

While I suspect these are fixable issues, these analyses are critical to the manuscript and it is a little disconcerting that they seem to be muddled, and thus need to be clarified.

4. Can the authors perform an analysis as to whether FoxG binding site motifs are also present in intron 1 of Wnt in other species? This would substantially support their claim of conservation of FoxG-induced stimulation of Wnt.

Minor points:

1. As the study is primarily based on chromatin and TF responses in regeneration, I think the manuscript would benefit from a more detailed exploration of the literature in these topics in the Introduction. Put another way, the current Intro. reads very nicely, but doesn't really set up the questions and motivation to look at chromatin responses in regeneration.

2. It would be nice to see an exemplary figure panel of the ATAC and ChIPmentation data in Figure 1 (“seeing is believing”).

3. I did not see a description of how many replicates were used in ATAC or ChIP experiments, although in the methods it states that “Twenty animals were used per biological replicate.” Perhaps I just missed it, but please add n = __ in either the main text (preferably) or appropriate methods section, so the reader can adequately interpret the reproducibility of the data.

4. The label in Figure panel 2C is a mess...

5. Figure 3a, the 'Reads coverage' data and ATAC-seq peaks don't seem to overlap for Enhancer 1, can this be explained?

6. In Figure 4b, what criteria was used for 'enhancer activity evidences'. Presumably some combination of chip/atac data, which is totally fine, but this should be explained in the methods section, and possibly figure legend. Also, 'evidences' should just be 'evidence'.

7. In figure 5a, 70% of foxG RNAi animals showed a tailless phenotype, 10% showed tails, and 20% showed... presumably two-headed?

8. I found the last results section "FoxG specifies posterior identity" a bit confusing at first, and think the manuscript would be strengthened if this section is re-written. This is a very exciting aspect of the work, which should be described as clearly as possible for full impact. First, I didn't understand on the first pass that these were cut and regenerated animals, which is a critical point of the experiment. Additionally:

"Eventually, some animals showed the strongest two-headed phenotype".

What does 'eventually' mean here, after some amount of time? If so, how much? How many animals exhibited this phenotype? Also, what does the 'strongest' two-headed phenotype mean, stronger than what, other phenotypes?

Moreover, there are references to Fig 4 that are clearly meant to be references to Fig 5.

9. The Introduction could also benefit from an elaboration of early vs. late Wnt expression, as this is an important distinction for phenotypic outcomes in RNAi experiments and interpretations of results.

10. Definitely not necessary for publication, but I think there are interesting connections to pioneer TFs and early developmental processes, and context could be included in the Intro. or discussion to broaden the interest of this work beyond planarian regeneration.

REVIEWER COMMENTS

Reviewer #1 (Remarks to the Author):

The submission from Pascual-Carreras et al integrates genome wide analyses that attempts to identify cis-regulatory elements during early regeneration in planarians, interpret this data in light of RNAi of key genes, and correlate this with later gene expression changes. They also provide the very interesting FoxG(RNAi) phenotype and attempt to link this to the regulation of *wnt1*, this data is relatively compelling. This a potentially very exciting and pioneering study for this model organism. Overall, in its current form the work is difficult to follow, there is little in terms of data quality control presented (although this may have been performed), and as a result the analysis of data requires some further work and hypothesis testing. There is no assessment of the the quality of the ATAC/ChIP data or the statistical significance of their results/analyses relative to any null hypothesis. While they use standard approaches and software there needs to be a more careful assessment of the significance of the data and more detail and clarity presented on how data was analysed. This issue needs to be addressed before this work can used as the basis of future work (and hence peer reviewed publication). They need to perform and present these QC analyses to demonstrate their findings, regarding CRE identification, motif enrichment and subsequent inferred regulatory interactions are potentially real. I should note that it seems likely they could be as there much of what they have found would be predicted by existing work in the literature. In order to help I have tried to outline what some of these QCs and analyses would below. I also have a number of comments, suggestions and questions about the data which I hope will be helpful.

Comments about experiments/data in the paper as presented.

1. Why did the authors only look at RNAs-seq at 0, 24, 48 and 72 hours of regeneration after *wnt1* RNAi. Why not use matched 12 hour samples for comparing to ATAC-seq experiments? It seems that this would be the most informative timepoint.

The RNA-seq experiment was designed to collect samples at 0, 12, 24, 48, 36 and 72 hours of regeneration (hR) in control and *wnt1* (RNAi) conditions. RNAi animals were treated using a mild inhibition strategy, aiming to obtain animals with a tailless (mild) phenotype, and not bi-headed (strong). Evaluating the *wnt1* expression in all the samples, we observed that one replicate of *wnt1* (RNAi) at 12hR and 36hR present the same *wnt1* expression levels than the controls (**Computational Supplementary Methods (CSM)***, section 3.4.4, Figure CSM.10), therefore we decided to exclude them. That could be the reason why differentially expressed genes (DEG) were found at 0, 24, 48 and 72 hR but not at 12 and 36 hR. This is shown in the volcano plots for those time points in the CSM, section 3.4.8 (Figure CSM.11). We could also notice that even *wnt1* was not a DEG, at 36h it was one of the main downregulated genes. In order to increase the chances to identify some differentially expressed genes at these time points, different statistical parameters were used (e.g. FDR cut-off=0.1) without any success. To summarize, even though the 12 hR time point was included in the design of the RNA-seq experiment and subsequently analyzed, we did not detect any DEG and therefore it was not useful to compare with the ATAC-seq data.

* In the revised version of the manuscript We have updated the "Data availability" section with two links to raw sequencing data and to the computational methods, including their code and relevant data files derived from those analyses. Sequencing datasets have been deposited with links to BioProject accession number PRJNA800775 in the NCBI BioProject database (<https://www.ncbi.nlm.nih.gov/bioproject/>). All the Computational Supplementary Methods (CSM) are available from GitHub (https://github.com/CompGenLabUB/2022_NatComm_BImethods). Those links will be publicly available upon publication, yet we provide the following temporary private links for the referees to access in the meantime: BioProject from NCBI submission portal (<https://dataview.ncbi.nlm.nih.gov/object/PRJNA800775?reviewer=th8qo8ld9ofvaj4a6euosiqape>) and GitHub repository through GitFront (<https://gitfront.io/r/JosepFAbril/uDS7i2Ym7fTE/2022-NatComm-BImethods/>).

2. Do the authors have ATAC-seq data for un-wounded tissue as a good control data set? This would allow them to set strict criteria for changes in accessibility.

We have clarified in the text that 0 hours of regeneration is considered as un-wounded. The tissue was dissected, dissociated, and fixed right away. However, this time point was not used to compare the accessible regions among the regenerative time points, since we only had one replicate. Instead, it was used to annotate promoters and enhancers over the whole genome as it is shown in CSM section 7.1. In addition, we have added ATAC-seq reads coverage from the 0 hR (un-wounded) and the 48 hR of anterior and posterior wounds in the genome browser (as an example, see Figure S7). Now, the reviewer and the community are able to explore those tracks.

3. The FoxG data with respect to establishing direct regulation and the potential broader role of FoxG is incomplete if the reader is to assess the evidence properly. What level is the H3K27ac signal at? Is this significant? Is ATAC data the same between both wounds or different? Are the peaks containing the FoxG motif called as significant by the methods presented in the paper? What other motifs are there in these enhancers? Given the significance of this for the authors manuscript/narrative I would suggest presenting the raw mapped read data for both signals, and providing a measure of their significance "how likely are they to be active enhancers.

Thanks for this comment, it is an interesting point. We have updated the tracks available from the genome browser, including the H3K27ac ChIP-seq read coverage from the 12 hR of anterior and posterior wounds, and, as mentioned before, the ATAC-seq read coverage from unwounded planarians (0 hR) and the anterior and posterior wounds at 48 hR. With these new tracks, the visualization has improved, being able to observe the presence of H3K27ac peaks in the *wnt1* first intron enhancer in Figure 3a. Additionally, a new Figure S7 has been added to show the accessibility and H3K27ac coverage of the first *wnt1* intron including 12 hR anterior wounds, 0 hR and 48 hR anterior and posterior wounds. Interestingly, we have noticed different accessibility of E1 and E2 during regeneration. Although we can not perform statistical analysis due to having just one replicate at 0 and 48 hR, it seems that while E2 is always accessible during the regeneration process, E1 is accessible just at 12 hR. We have added this new information in the main text and further discussed it.

We also have explored the presence of other TF binding motifs in enhancers 1 and 2. The enhancers 1 and 2 contain 78 and 51 TF motifs, respectively (Table S6). The reviewer and

the rest of the community could also explore the presence of other TF binding site motifs in the genome (<https://compngen.bio.ub.edu/jbrowse>) using the appropriate tracks and the tool developed for this specific purpose (https://compngen.bio.ub.edu/PlanNET/tf_tools). Further details about the motifs found on the predicted *wnt1* intron enhancers can be found on Figure CSM.41 (GFF files for the annotated motifs are also available through the supplementary material git page).

Overall, we have presented and made accessible all the raw data generated for this study, as the reviewer suggested. We have also further studied the chromatin accessibility and the TF presence of two CREs located in the first intron of *wnt1*, suggesting that FoxG could have a role in the regulation of *wnt1* expression.

4. Do ACRs/Enhancers identified in the first intron of Wnt1 of other species also contain FoxG binding motifs? This is not clear in the paper as presented, this should be shown. Does the evidence for ACR and enhancer in the first intron overlap (along with a potential binding site?)

This is a very good observation. We have identified putative FoxG binding motifs in the ACR and enhancers in the first intron of *wnt1* from most studied species, such as *Danio rerio*, *Xenopus tropicalis*, *Drosophila melanogaster*, *Hofstenia miamia* and *Hydra vulgaris* (Figure 3c, Table S7). In all the species, the potential FoxG binding site was located in a region with ACR and/or enhancer evidence. Interestingly, we did not find it in mammals. A new methods section was added to explain the analysis. Furthermore, the possible evolutionary conservation on the regulatory regions present in the first intron of *wnt1* has been discussed.

5. What other potential FoxG targets are implicated by this simple method, and are they also wound induced genes?

We have explored the presence of the FoxG motif in the CREs of the wound-induced genes (Wurtel et al. 2015), finding 28 wound-induced genes with FoxG in their CREs (Table S8). We have also studied the presence of the FoxG motif in the specific anterior and posterior promoter and enhancer regions annotated in this study (a list of these genes can be found in the Table S9). To study the function of the genes associated with those CREs containing the FOXG motif, we have performed a gene ontology analysis. The results can be found now in Figure S10 and Table S9 (CSM section 7.4).

6. The FoxG expression pattern seems to disperse as presented, is there expression in cell at the anterior margin?

We have improved the ISH for *foxG* showing that its expression is all around the body, also in the anterior margin. Importantly, we now show that *foxG* is expressed in scattered cells in 24h wound regions (Figure 4d), as already shown with *wnt1* expression, supporting its direct regulation. Furthermore, we now show the expression of *foxG* in the CNS and in the blastemas (Figure S8), suggesting that it could also have a role in the differentiation of neurons, as described in other models (Kumamoto and Hanashima. *Dev. Growth Differ.* 59, 258–269 (2017)).

7. Would the authors be able to look at ATAC signal in FoxG(RNAi) animals to see if the signal changes? Generally, Fox family TFs are classed as pioneers.

Although it would be very informative to study the accessible chromatin regions in *foxG* (RNAi) animals and therefore understand its putative role as a pioneer factor, this experiment is not easily affordable. We thank the reviewer for pointing this out, and we will consider including it in the next analysis.

Comments on the quality control of data and data analysis approaches.

ATAC-seq Quality control.

1. Given this is one of the first, and the first extensive examples of ATAC-seq on planarians the authors need to supply more quality metrics for ATAC-seq, including tagmented library profiles and measures of the proportion of mapped reads under peaks. This will allow readers to assess the quality of libraries, and the level of tagmentation in these libraries. They also need to show the global level of peaks around their center of highest coverage and describe the distribution of peak widths, showing which are significant (called as peaks). They should also present the ATAC accessibility around predicted TSSs across the genome. This will establish the ATAC data is reasonable, not over tagmented (although this can be useful too).

We thank the reviewer for pointing this aspect out. We had already run ATAC-seq and ChIPmentation quality assessments; initially those analyses were done using ATAC-seq QC (now included in CSM sections 4.2.4 and 5.2.3, for ATAC-seq and ChIPmentation respectively). We took the opportunity to improve the visualization using deepTools, more recent and customizable, for the reviewed version of the manuscript (and included that in CSM sections 4.3.2 and 5.3.2). Peak-centered heatmaps across different genomic locations for the ATAC-seq merged replicates are shown in Figure CSM.19, and also shown separately by replicate in Figure CSM.81. We also provide heat maps centered over the transcription start sites (TSS) on Figures CSM.76, CSM.77, and CSM.78, where one can spot the expected pattern of reads mapping on the TSS regions for the nucleosome-free (NF) peaks. Library complexity and fragment size distribution plots for all the replicates are also available in the CSM document. Several quality measures were also computed—including Promoter/Transcript body (PT), Nucleosome Free Regions (NFR), and the Transcription Start Site Enrichment (TSSE) scores—, along with the corresponding plots (for those see CSM section 9.3.3). It is worth saying that all replicates of all samples had a TSSE score above the recommended ENCODE protocols threshold for non-model organisms (other than human, mouse and flies), which was defined to be larger than 7; on Table CSM.T7 most of the values are above that threshold. It is also worth taking into account that *S.mediterranea* genome version (Grohme et al 2018) being used was far from finished and that we have chosen as reference gene annotation the “SMESG-repeat filtered” to include as many predicted genes as possible (curation quality for the planarian annotation cannot be compared with human, mouse or fly either).

2. The authors should present biological replicate and experimental condition correlation plots to look at the global congruence between replicate peak calling, to get more assurance that the approach is working well.

We applied a replicate correlation analysis for RNA-seq replicates (see CSM section 9.2.2); the correlation among ATAC-seq samples was somehow considered by DiffBind when looking for the significant peaks across those conditions. On this item, we think that the referee here meant a standard measure implemented for this purpose known as the Irreproducible Discovery Rate (IDR). Pair-wise comparison of the replicates is provided on Figure CSM.82, where one can find the rank plots of all those comparisons showing high concordance among replicates. Numbers included on the Venn diagrams of Figure CSM.83 also show large amounts of peaks shared among replicates.

3. Also they should include a comparison to previously published ATAC data as a QC step (Gehrke et al, 2019), some of the accessible regions should be the same.

This analysis could be useful as a QC. However, from the referred manuscript (Gehrke et al, 2019, <https://doi.org/10.1126/science.aau6173>) we were able to access only the raw reads for the planarian ATAC-seq (<https://www.ncbi.nlm.nih.gov/bioproject/?term=PRJNA515075>), not the BED/GFF files for the already annotated peaks for that experiment; Furthermore, script templates for that reference were provided (<https://zenodo.org/record/2547750#.YwjnNNJBxhE>), yet not the real scripts run for the analysis; which means that, if we had to, we were not able to exactly reproduce all the analyses and to retrieve the regions described in that manuscript for comparison against our regions.

Although it could have been useful, we consider that having replicates at 12hR is a much better QC than comparing with an analysis performed at a different time point (6hR).

4. The authors present data of ATAC peaks that show changes in accessibility after RNAi. These numbers suggest changes in most of the identified posterior or anterior specific CREs. I am not clear on how these changes in accessibility were measured or assessed? What were the criteria for these classes? The authors need to explain this and present some specific examples in the paper. They should also clarify the overlap in these changes with Venn diagrams as we might expect them to change in opposite directions for some peaks?

Thanks to the reviewer's comments. We have included a methods section, carefully explaining the criteria followed to design the accessibility of the chromatin regions (Accessible, slightly accessible, less accessible, non-accessible). Moreover, we have added an example plot for each condition to better visualize and clarify how each mentioned accessible chromatin region behaves in *wnt1* and *notum* (RNAi) conditions (Figures S2, S3 and S4). The computational analyses and the thresholds assigned to determine the accessibility status of the CREs are described in CSM section 5.4.2.

5. The authors refer to previous ATAC and and Chipmentation samples (0-48 hours) what are these precisely? Is there a reference for them?

We have not used any previously generated ATAC or ChIP samples. We have generated all the ATAC-seq and ChIPmentation samples in this study. We have clarified in the text that an ATAC-seq generated for un-wounded planarians (0 hR) and the anterior and posterior wounds at 48 hR was also generated. As there was only one replicate for each of those three samples we were not able to integrate them with the 12hR samples, yet they were still useful for the

ChIPmentation Quality Control

Similar to the ATAC-data, the authors need to show controls and QCs for this approach. There are a number of possibilities.

1. They could include comparison to other available data sets, even if the H3K27ac is not available there are other marks that they could look at for correlation.

ChIPmentation analysis of H3K4me3, H3K4me1, H3K27me3 and H3K36me3 have been performed in planarians (Mihaylova et al. 2018, Dattani et al. 2018). However, these analyses were performed in X1 FACS-sorted cells, which is in a stem cell enriched *in vitro* context totally different to the one analyzed in the present study, whose significance is its *in vivo* nature.

2. The correlation between these data and ATAC data should be considered globally, how many peaks do not overlap, is the overlap greater than that expected by chance?

Peaks computed by MACS2 for both ATAC-seq and ChIP-seq samples were compared by DiffBind, using similar parameters as for the ATAC-Seq analysis, which also calculates an adjusted p-value (FDR) to assign significance to the overlapping peaks. The intersection of ATAC-seq and ChIP data is described on CSM section 5.4 (see Figure CSM.28); moreover, summary plots for the corresponding DiffBind analyses are provided on CSM section 9.5.2. As the ChIP-seq peaks are compared against the significant ATAC-seq peaks, there are only peaks in the all-sets group of the Venn diagrams of Figures CSM.92 and CSM.93, that match to the 611 and 2484 peaks that were found significant for anterior and posterior peaks, respectively.

3. Are ATAC peaks bounded by H3K27ac peaks? (the authors are aware of this pattern as they use it later in the paper).

To define the enhancer regions we have merged the ATAC-seq and ChIP-seq data sets in order to find overlaps that were then assessed by DiffBind, as described on the previous answer, resulting in genomics segments where one can find coverage for both, ATAC-seq and ChIP-seq, or only for one of them. Looking at the peak-centered heatmaps on the CSM Figures CSM.17 and CSM.23, one can see that the coverage in the enhancer clusters (enh.DWN and enh.notDWN) with respect to the other categories switches when we are looking at ChIP-seq data (stronger signal on those clusters in Figure CSM.23), and ATAC (stronger signal on the other five clusters in Figure CSM.17). However, one can see some coverage signal in the exon/intron clusters, apart from the intergenic one in figure CSM.23.

4. Is their correlation with, for example, H3K4me1 signal? These data sets do exist in some form for planarians and are available in the short-read archive.

As exposed in point 1 of this section, we consider that the analysis performed in sorted G2/M cells of dissociated tissues is not comparable to our analysis, corresponding to *in vivo* tissue of posterior wound regions, which present a higher cellular complexity. It could be interesting to analyze, but we do not think that it could be considered as a QC.

Calling CREs and potential Transcription factors at 12 hours, and then at later timepoints.

The authors need to query their data for noise to signal. For example, looking for motifs in random blocks of sequence amongst the same genomic regions around differentially expressed genes vs all genes, or just in ATAC peaks or H3K27ac peaks. The second set of CREs discovered using ATAC at different timepoints is not supported by chipmentation data as far as I can ascertain, so again it is very important these data are backed up by rigorous QC and some analysis to show that motif enrichment under peaks is distinct from motif enrichment in the same region with low/zero ATAC coverage. These later ATAC data also need standard QC as described above for the 12-hour timepoints.

To illustrate this can the authors show the presence of potential TCF sites around potential target genes to see if they are actually concentrated under peaks, look at peaks in genes that are expressed but not differentially and show they don't have TCF site enrichment, similarly for genes that are up-regulated what is the collective presence or absence of TCF sites? Without these analyses the authors only look for what they wish to see and have no idea if it is meaningful or just chance occurrence. For the homeobox analysis TCF sites also need to be investigated in the homeoboxes that do not appear to be regulated with respect to genes expression analysis- these should not have TCF sites.

The motif enrichment analysis should also be presented for up-regulated genes as a comparison.

To make the data useful somewhere all the planarians orthologs to the transcription factors presented should be presented for the reader (where they actually exist, i.e. not Nanog!)

Quality control has been undertaken over all the sample replicates for all time points of ATAC and ChIP data. Once the CREs have been defined, Homer calculates if a motif is overrepresented in those regions with respect to either a set of random segments taken from the genome, or a set of regions provided by the user (for instance when looking for motifs on CREs associated to down-regulated genes, the CREs for the not down-regulated genes were considered as background). The chosen background depends on the analysis undertaken, yet it can be found when each has been applied to the corresponding code blocks of CSM sections 6 and 7. Thus, our analysis indicated the presence of TCF binding sites in the regulatory regions analyzed. The aim of this analysis was not to conclude that the down-regulated genes after *wnt1* RNAi had enrichment of TCF binding sites, but to directly analyze which of the down-regulated genes showed TCF binding sites and could be direct targets of *wnt1*. TCF binding sites can be found in genes regulated by the canonical *wnt* signal (mediated by beta-catenin), independently of the *Wnt* that is activating it. For this reason, the aim was not to find an enrichment of TCF binding sites in this group of genes, since anterior genes are also regulated by the canonical *Wnt* pathway. We have now rewritten the main text and avoided the term overrepresentation.

Regarding the last comment, there are no TF binding site databases in planarians, so the binding site motif calling was performed against Human/*Drosophila* databases. For this reason, in Figure 2C we have included the motif names defined on the Human/*Drosophila* TF

databases. Annotating the planarians' orthologs for those potential TF binding sites would generate inaccuracy.

Minor comments on the clarity.

The paper could do with significant editing for clarity and grammar to help the reader.

Abstract

The abstract could be re-written to more clearly reflect the results. To my knowledge “wound cells” is not a working definition of any cell type or state. Do the authors mean that “amongst cells at the wound site chromatin accessibility and histone marks change between anterior and posterior facing wounds”. This also appears in the introduction line 84, and could be clearer for non experts.

The authors study chromatin confirmation by the normal definition as direct contacts between enhancers/promoters etc are not measured using a Hi-C type approach. Through the paper conformation should probably be changed to accessibility or ATAC-signal.

“pre-existent” should be “pre-existing”

“WNT/beta-catenin signal” should be “signalling”

I would advise that the final point regarding FoxG is worked into the main body of the abstract as this the major implication of this of this work rather than just a “Furthermore,”

Similar clarity issues exist through the manuscript that could be improved to help the reader before any subsequent submission.

Thanks, every minor comment has been considered for the modified manuscript.

Introduction

Probably pertinent to mention the recent description of activin-2 as being required for the asymmetric expression of Notum (ref) (line 72) i.e. something is now known about this.

We thank the reviewer for suggesting this reference, we decided to add it in the discussion section.

Results

See above, but clarity of message could be improved for the reader with careful editing.

Thanks, we have tried to improve the clarity of the message.

The discussion is well written and comprehensive, if rather selective in mentioning transcription factors that control anterior and posterior regeneration.

The transcription factors that control posterior regeneration are included in the discussion. However, we have not included the ones that control anterior, since our study focuses in the posterior region, and we think that adding so much data would compromise the final readability of the arguments in the discussion.

--

Reviewer #2 (Remarks to the Author):

In this manuscript, the authors explore polarity reestablishment during planarian regeneration with a focus on how chromatin reorganization might play a key role. The first half of this paper is a tour de force of methods that include ATAC-Seq and ChIPmentation, which are new or nearly new to the planarian community. The authors also use RNAi/RNA-Seq and site prediction strategies to connect their chromatin discoveries to Wnt/Beta-catenin signaling and the cellular/molecular basis of reestablishment of anteroposterior polarity. In the second half of the paper, the group discovers a transcription factor, FoxG, that is important for expression of *Wnt-1*, a key signaling molecule in the A/P signaling process. The mechanism for this regulation appears to be well-conserved. There are many aspects of this work that are novel and exciting and the use of new techniques is innovative, as well. However, the two halves of the paper are not clearly linked. Further, the work requires several major changes or additions before it would be appropriate for publication in Nature Communications.

Major issues:

1) The most difficult flaw in this paper is that the two halves of the paper are not well connected, either in writing or conceptually. The first half of the paper seeks to explore changes in chromatin status that occur as a result of Wnt/Beta-catenin signaling perturbation, but then the second half of the paper swings around to explore how *Wnt1* is itself controlled by FoxG, a series of results that are interesting but fully unconnected to the chromatin reorganization story. One solution to this problem would be to separate this paper into two; this would depend on the wishes of the authors and editors. Alternatively, the authors would need to rework the manuscript or add data so that the manuscript works well together.

We believe that the two parts of the paper are connected enough since we identified foxG thanks to the genomic analysis. In any case, we have tried to strengthen this connection by providing new data about FoxG. Now we show that the E1 of *wnt1* seems to be regeneration-specific (Figure S7), and that there is the presence of foxG binding sites in the first intron of *wnt1* in different species (Figure 3C). Furthermore, we demonstrate that foxG is expressed in a salt-and-pepper manner in 1dR blastemas, as found with *wnt1* expression (Figure 4d). In this revised version of the manuscript, we have also reorganized the data in order to improve the connection between the genomic and the functional parts of the study.

2) The title indicates that “Wnt/beta-catenin signal controls pole-specific chromatin remodeling during planarian regeneration.” The “control” element of this claim is not well supported. The authors do demonstrate a correlation between chromatin changes and Wnt/Notum perturbation, but the mechanism connecting Wnt and chromatin is not at all explored.

a. The authors claim that some chromatin remodeling factors are downregulated after Wnt(RNAi), but the fold changes for the two genes mentioned are modest and no functional data are included to show that BPTF and TOP2B have any role in chromatin organization changes seen after Wnt perturbation and/or polarity of the planarian body. Identification of the Wnt-regulated factor that causes A/P chromatin changes would be ideal.

b. Though chromatin remodeling or modifying enzymes might be targets of Wnt signaling, another possibility is transcription factors downstream of Wnt signaling (either beta-catenin/TCF? or TFs differentially expressed downstream of Wnt) act as pioneer TFs to recruit

chromatin modifying/remodeling enzymes to particular regions of the genome. This possibility could also be explored bioinformatically and experimentally.

The reviewer is right. However, we have found that chromatin remodelers as BPTF and TOP2B are downregulated after *wnt1* RNAi and have TCF binding sites in their CREs (Table S2-5), which agrees with reported data in different species and development contexts supporting that *wnt/bcat* signaling controls chromatin remodeling.

In any case, we agree that our results do not clarify whether it is mainly a direct role or whether it could activate pioneer TFs that recruit chromatin-modifying/remodeling enzymes. We have specified it in the text and we have modified the title of the manuscript to “Wnt/beta-catenin signaling is required for pole-specific chromatin remodeling during planarian regeneration”.

3) The methods are not well-described, in terms of technique or data analysis. This needs to be improved for publication, particularly since these are newer methods for planarians and some of this paper's impact comes from the establishment/adaptation of new methods. The citations provided are not for planarian samples. Information about controls or evaluation of quality of data are not included. In addition to a general improvement in the methods section, the following are important both for reproducibility and interpretation of data.

Thanks. We have improved the methods section, clarifying the methodology used to obtain the samples and the quality controls. In the revised version of the manuscript We have updated the "Data availability" section with two links to raw sequencing data and to the computational methods, including their code and relevant data files derived from those analyses. Sequencing datasets have been deposited with links to BioProject accession number PRJNA800775 in the NCBI BioProject database (<https://www.ncbi.nlm.nih.gov/bioproject/>). All the Computational Supplementary Methods (CSM) are available from GitHub (https://github.com/CompGenLabUB/2022_NatComm_BImethods). Those links will be publicly available upon publication, yet we provide the following temporary private links for the referees to access in the meantime: BioProject from NCBI submission portal (<https://dataview.ncbi.nlm.nih.gov/object/PRJNA800775?reviewer=th8qo8ld9ofvaj4a6euosiqape>) and GitHub repository through GitFront (<https://gitfront.io/r/JosepFAbril/uDS7i2Ym7fTE/2022-NatComm-BImethods/>).

a. Because planarian promoters are not at all well characterized, more discussion should be included about how peak calling was made, how peaks were assigned to nearby genes (for example when peaks are near more than one gene), what controls were used, how proximal/distal promoter definitions were defined, etc. A general discussion about trends seen for planarian promoters (in the text or in the supplement) might also be warranted, as this information could be more broadly valuable to others trying to understand gene regulation and promoter structure in planarians.

As it has been answered to reviewer 1, peaks were computed using MACS2, filtering for both ATAC and ChIP samples; then those were compared by DiffBind, using similar parameters, which also calculates an adjusted p-value (FDR) to assign significance to the overlapping peaks. The intersection of ATAC and ChIP data is described on CSM section 5.4 (see Figure CSM.28); moreover, summary plots for the corresponding DiffBind analyses are provided on CSM section 9.5.2. As the ChIP peaks are compared against the significant ATAC peaks,

there are only peaks in the all-sets group of the Venn diagrams of Figures CSM.92 and CSM.93, that match to the 611 and 2484 peaks that were found significant for anterior and posterior peaks respectively. Taking into account the reference gene annotation from PlanMine (SMESG-repeat, see Table CSM.T2), Homer annotatePeaks script was used to classify promoter peaks into 1549 proximal and 2594 core promoter regions, as shown on CSM section 7.2.1. Same approach was taken to classify enhancer regions; 28720 distal enhancer, 19610 proximal enhancer, and 3157 first-intron enhancer motifs were defined after that procedure, as shown on CSM section 7.3.1.

b. ChIPmentation methods should be extensively detailed, since this is the first use of this method in planarians (to my knowledge). The H3K27ac antibody used should be stated. Controls for this particular experiment should be explained.

Thanks for the comment. We have improved the methods section, explaining in detail the ChIPmentation method for planarians. In addition, a western blot gel image has also been included in the computational supplementary methods demonstrating that it recognized a protein with an expected size in planarian protein extract (CSM, section 5, Figure CSM.24).

c. The authors should clarify to the reader what criteria were met for a region to be called an “enhancer,” as this term has a particular functional connotation and function of these sequences have not yet been studied. The authors could consider using a less functional term “accessible region” until function is determined.

We have clarified the criteria of promoter and enhancer definition and the Methods section. Additionally, we have used the term “putative” when describing the genomic regions annotated in this study.

d. Further, how distinctions for enhancers like “accessible/slightly accessible/less accessible/non-accessible” were made warrant clearer definition. What are thresholds for these different categories?

Thanks to the reviewer’s comments. We have included a methods section, carefully explaining the criteria followed to design the accessibility of the chromatin regions (Accessible, slightly accessible, less accessible, non-accessible). Moreover, we have added an example plot for each condition to better visualize and clarify how each mentioned accessible chromatin region behaves in *wnt1* and *notum* (RNAi) conditions. We show it in Figure S2, S3 and S4, and further details are provided on CSM section 5.4.

e. The ATAC-seq analysis of 0 and 48 h wounds are referenced, but only briefly. This may also warrant further explanation. It isn’t clear why these analyses were included and how these results were integrated with other results.

We have clarified that 0 hours of regeneration is considered as un-wounded. The tissue was dissected, dissociated, and fixed right away. However, this time point was not used to compare the accessible regions among the regenerative time points, since we only had one replicate. Same applies to the anterior and posterior samples at 48hR. As we already explained to

referee 1, there was only one replicate for each of those three samples. Therefore, we were not able to integrate them with the 12hR samples, yet they were still useful for the whole-genome CREs annotation described on CSM section 7. In addition, we have added ATAC-seq reads coverage from the 0 hR (un-wounded) and the 48 hR of anterior and posterior wounds in the genome browser (as an example, see Figure S7). Now, the reviewer and the community are able to explore those tracks.

f. The authors should clarify to the reader which approaches are novel and which have been used for planarians before. This will help the reader appreciate the novelty of this work, which might be overlooked otherwise. Where other similar approaches have been used, adding more clarity will also give appropriate credit to other related work and help put this manuscript in context.

We thank the reviewer's comment. We have added more information on this behalf in the discussion to provide more context to the readers.

4) It bears clarifying that the transcription factor binding site enrichment approach is based on the binding sites of TFs from other organisms. The TF binding sites for planarian TFs have not been determined. Therefore, some results from this approach may well indicate conserved binding sites, but others could be misleading. Care should be taken to avoid overinterpretation based on these binding sites without experimental validation.

a. One way to connect the two stories would be to actually characterize the accessible elements and binding sites for wnt 1, experimentally verify the FoxG binding site, and assay whether FoxG does bind to the wnt1 promoter and elsewhere.

We are aware that TF binding sites in planarians have not been determined, so the binding site motif calling was performed against Human/Drosophila databases. We do not aim to compare the potential conserved binding sites between these species and planarians but to use them to predict the potential binding sites of the homologous TFs in planarians.

We have tried to perform a luciferase assay to test the activity of the putative wnt1 enhancer 2 and its activation by FoxG (see Figure below). According to the luciferase quantification, the enhancer alone transfected in HeLa cells is already active (Orange bar), and Smed-FoxG does not enhance in a significant manner the activity of the putative enhancer 2 with respect to the enhancer alone (yellow bar). This result suggests that this putative enhancer identified in our analysis does have enhancer activity. However, in this assay we cannot demonstrate that it is activated by FoxG. We performed the same experiment in Saos cells, and the result was very similar (not shown). Our results are not conclusive and several controls and further analysis are missing. For instance, the expression of smed-FoxG and the endogenous FoxG should be analyzed. That is the reason why this essay has not been included in this version of the manuscript, but we plan to continue with this functional analysis to understand the mechanism of wnt1 transcriptional activation.

Furthermore, while conducting these experiments a manuscript identifying promoter and enhancer regions in the Smed genome was published in eLife (Neiro et al.11, 2022.02.03.479047 (2022)). Interestingly, in this study the authors also find an enhancer-like region in the first intron of wnt1 with a high level of H3K27ac, H3K4me1 and ATAC-seq

footprinting scores for Fox family transcription factors, which corresponds to the putative Enhancer 1 found in our study (this data is now included and discussed in the revised version of the manuscript). Additionally, our analysis indicated that E1 could be specifically accessible during regeneration (Fig S7). Thus, we plan in the next study to perform the luciferase assay also with this Enhancer 1.

Graphic showing the luciferase activity in HeLa cells transfected with a plasmid containing the putative Enhancer 2 of *wnt1* +/- an expression vector containing *smad-FoxG*.

5) A few questions about FoxG remain:

a. foxG expression in Fig. 3d appears to be fairly ubiquitous. Can the authors explain how FoxG promotes *Wnt1* expression in such a specific location despite FoxG being present elsewhere? Where/when is foxG expressed in regeneration? Is it also coexpressed in anterior *wnt1*+ cells?

Thanks, this is an important point that we have now improved. Using WISH, we have explored the expression pattern of *foxG* during regeneration. Interestingly, at 24 hR *foxG* presents a salt-and-pepper expression pattern in the anterior and posterior blastemas, resembling the expression of *wnt1* in this exact time point, and thus supporting its function in regulating *wnt1* expression. We show it now in Figure 4d.

We have also explored the expression of *foxG* in a late regenerating time point, showing a broad expression in anterior and posterior blastemas at 72 hR (Figure S8). At this time point, its expression in the CNS is evident, thus suggesting that FoxG has also a role in the differentiation of the nervous system, as in other models (Kumamoto and Hanashima. *Dev. Growth Differ.* 59, 258–269 (2017)). This information is now included in the results and discussion.

Although we have demonstrated *in silico* that foxG and *wnt1* are present in the same cell types (Figure S9a), we have also tried to perform FISH to demonstrate the coexpression of *foxG* and *wnt1* during regeneration. However, this was not technically possible.

Thus, our evidence supports the role of FoxG as an activator of early *wnt1* expression. Additionally, we have further discussed the idea that the presence of foxG is not the only limiting aspect of *wnt1* expression. As mentioned, we have discovered that E1 seems to be specifically accessible during regeneration (Fig S7), suggesting that it could be context-specific. As a consequence, the presence of foxG in *wnt1* cells would not specifically promote its expression. In the same way, E1 and E2 present different TFs binding motifs, indicating that other TFs or cofactors could be essential for *wnt1* expression. We have added this information to the manuscript and discussed it.

b. Are the predicted FoxG binding sites (in *wnt1* or elsewhere) differentially accessible in posterior samples from the ATAC-seq data?

We have explored the presence of FOXG binding sites in the anterior and posterior CREs annotated in this study. Interestingly, we found a similar number of genes associated to CREs containing a foxG binding site corresponding to A or to P regions.

Furthermore, we found that the E1 (containing a FOXG binding site) of *wnt1* seems specific for regeneration but it is open in A and in P regions of 12h regenerating animals, which in fact agrees with the expression of *wnt1*.

c. The foxG(RNAi) experiments in Fig. 5b (especially *hox4b* and *sp5*) are not always clear would be more quantifiable (with statistical analysis) with RT-qPCR.

To better demonstrate that in foxG (RNAi) animals the expression of posterior markers was significantly reduced, we have performed qRT-PCR of the same markers (*fz4*, *post2d*, *hox4b* and *sp5*) in control and foxG (RNAi) animals, showing the expression of all four markers was reduced, being able to perform statistical analysis. We have added this data in Figure S9d.

Minor issues:

1) N for several experiments is needed (in figure legends, for example in 5A, 5C). N should be added throughout.

Thanks for mentioning this aspect. We have added the N in Figure 5a. Additionally, we have also clarified that we have obtained the same phenotype proportions in two independent experiments. The Figure 5c is a magnification of one animal obtained in the same experiment.

2) ATAC-Seq or similar methods performed on complex tissues probably misses some cell type-specific complexity in chromatin arrangement. It could be useful to discuss limitations of these approaches in the discussion. For example, are “anterior” or “posterior” specific peaks expected to be the same in all cells or changing only in specific cell types (e.g. muscle)?

We have included a paragraph in the discussion highlighting the requirement of SC-genomic approaches to uncover cell-type specific changes.

3) In Fig. 1A, it isn't clear how many peaks were in both A & P samples.

Those numbers can be retrieved from the Venn diagram on Figure CSM.91. We provide all those numbers in CSM section 9.5 for other DiffBind comparisons. Moreover, from all the shared peaks for the anterior/posterior comparison, 3370 were found significant (FDR<0.05)

and of those 611 and 2484, for anterior and posterior respectively, had an absolute fold change > 2, as it is shown in Figure CSM.27. After searching for differentially bound sites between those sets and anterior/posterior ChIP peaks, the peaks associated to putative enhancer regions were filtered out, 555 and 1869 respectively (see Figure CSM.28).

4) Figure 4 is fairly confusing and the binding sites and conservation of exons/binding sites could be more clearly demonstrated. I think my confusion arises because the intron is removed in A but is the focus of the conservation discussion.

Thanks for the reviewer's observation. We have improved this figures (now Figure 3b) to better show that the position of intron 1 in *wnt1* is evolutionarily conserved. We have simplified the B part, drawing boxes to summarize the experimental evidence for ACR, enhancer activity, and foxG binding site motif in the first intron of *wnt1* across species.

5) The authors state in the introduction that the existence of organizing centers in adults “has not been well studied,” but this is fairly well studied in planarians. Consider rephrasing?

We have rephrased the sentence to make it clear that we meant that organizing centers had not been well studied in adult tissue compared to embryos.

6) The plural of motifs is usually with an F.

Thanks, we have corrected it.

--

Reviewer #3 (Remarks to the Author):

The manuscript entitled “Wnt/ β -catenin signal controls pole-specific chromatin remodeling during planarian regeneration” by Pascual-Carreras et al., is a timely and exciting contribution to the field of regeneration and chromatin biology. Although not directly addressed, the manuscript also raises intriguing questions about the role of ‘pioneer’ transcription factors in establishing polarity during regeneration.

The Key Results of the manuscript are (1) that wnt and notum, which are known to be required for establishing polarity during regeneration in planarians, are required for chromatin remodeling and enhancer accessibility early after injury (<12 hours) well before the 1st anterior and posterior markers appear; (2) that the transcription factor FoxG is required for wnt expression in posterior and anterior wounds and for proper polarity after injury; and (3) that intron 1 of wnt bearing chromatin accessible regions with enhancer activity is evolutionarily conserved.

Broadly speaking, the approach and experimental design are sound; however I have a few major and minor points regarding additional analyses, data clarity and reproducibility, and suggestions for providing appropriate context in the text, that I believe would substantially improve the manuscript and should be addressed before acceptance. It is my belief that these are not extraordinary requests and that the authors can complete them in a timely manner so that the community can appreciate this important contribution.

Major points:

1. The manuscript would significantly benefit from a t'0 enhancer accessibility time point (prior to or immediately after injury). Are the enhancers identified in this manuscript specific to establishing polarity in regeneration, or do they just provide homeostatic A and P ends? It seems that such data might already exist and be publicly available? I highlight a sentence from their summary of chromatin accessibility (and the title of Figure 1) to emphasize this point:

“the chromatin structure of cells in the wound is remodeled according to the polarity of the pre-existing tissue, which occurs a few hours after amputation (<12 hR), before the first anterior or posterior markers appear (around 48 hR).”

Is it really remodeled in response to injury, or was it already there?

*Although it may not be this simple, the authors might also find that the FoxG enhancer in intron 1 of Wnt is closed prior to injury, and that the opening allows FoxG to bind, induce Wnt expression, and initiate posterior regeneration, which I believe would add an additional layer to their manuscript.

We thank the reviewer's comment. It is a really interesting observation.

In the main text, it is mentioned that the 0 hours of regeneration sample is considered as un-wounded. The tissue was dissected, dissociated and fixed right away. We have clarified this aspect in the text.

However, this time point was not used to compare the accessible regions among the regenerative time points, since we only had one replicate. Instead, it was used as a replica to annotate promoters and enhancers genome-wide. In the same way, we also performed ATAC-seq for anterior and posterior wounds at 48 hR, which was also used to annotate promoters and enhancers. We have clarified this aspect in the material and method section.

We have added ATAC-seq reads coverage from the 0 hR (unwounded) and the 48 hR of anterior and posterior wounds in the genome browser. Now, the reviewer and the community are able to explore those tracks. Interestingly, we have noticed different accessibility of *wnt1* E1 and E2 during regeneration. Although we can not perform statistical analysis due to having just one replicate at 0 and 48 hR, it seems that E1 is accessible just at 12 hR, and E2 is always accessible. This means that E1 is closed prior to injury and that its opening could be related to FoxG binding and *Wnt1* expression, to induce posterior regeneration, as the reviewer suggested. We have added this new information in the main text and Figure S7. We have also discussed this new revealing aspect.

2. There is not sufficient description of the bioinformatic methodology for assigning the different levels of chromatin accessibility in Figure 1. What distinguishes 'Accessible' from 'Slightly' and 'Less' accessible (also there is a typo in the figure legend that reads 'Slighly' accessible). As this is a critical component of the manuscript, this needs to be addressed.

1b. I know it's simple, but I think the manuscript would benefit from a brief description of the pole-specific filter to establish A- and P-specific enhancers in the methods section.

Thanks to the reviewer's comment. We have included a methods section, carefully explaining the criteria followed to design the accessibility of the chromatin regions (Accessible, slightly accessible, less accessible, non-accessible). Moreover, we have added an example plot for each condition to better visualize and clarify how each mentioned accessible chromatin region behaves in *wnt1* and *notum* (RNAi) conditions (Figure S2, S3 and S4), and further details are provided on Computational Supplementary Methods (CSM)* section 5.4.

* In the revised version of the manuscript we have linked the Computational Supplementary Methods, to which the reviewers can have direct access through the following link: https://compgen.bio.ub.edu/datasets/2022_NatComm/2022_NatComm_BImethods.pdf

3. Related to point #1; there seems to be a discrepancy between the reported enhancer accessibility and the Figure 1B panel. First, again, it is unclear what slightly vs. less accessible means. Second, the authors state that:

"in *notum* (RNAi) anterior wounds, only 12.3% of the anterior putative active enhancers were open, whereas the remainder were closed or reduced their accessibility",

however the pie chart in Figure 1B shows >25% accessible for this sample. Either the pie chart is only representative, in which case it should be re-made to reflect the data, or there is a discrepancy between the data and figure panel.

Third, the authors state:

“Furthermore, 31.4% of the anterior putative active enhancers appeared to be open in *wnt1* (RNAi) posterior wounds and 9.5% became more accessible”,

however the pie chart shows maybe 31.4% as “less accessible” rather than “accessible”, and maybe 9.5% as “accessible” rather than “more (?) accessible”.

While I suspect these are fixable issues, these analyses are critical to the manuscript and it is a little disconcerting that they seem to be muddled, and thus need to be clarified.

Thanks to the reviewer’s observation. We have fixed the pie charts, and now the manuscript fits the plots observed in Figure 1b.

4. Can the authors perform an analysis as to whether FoxG binding site motifs are also present in intron 1 of *Wnt* in other species? This would substantially support their claim of conservation of FoxG-induced stimulation of *Wnt*.

This is a very good observation. We have identified putative FoxG binding motifs in the ACR and enhancers in the first intron of *wnt1* from most studied species, such as *Danio rerio*, *Xenopus tropicalis*, *Drosophila melanogaster*, *Hofstenia miamia* and *Hydra vulgaris* (Figure 3b, Table S7). Interestingly, we did not find it in mammals. A new methods section was added to explain the analysis.

Minor points:

1. As the study is primarily based on chromatin and TF responses in regeneration, I think the manuscript would benefit from a more detailed exploration of the literature in these topics in the Introduction. Put another way, the current Intro. reads very nicely, but doesn’t really set up the questions and motivation to look at chromatin responses in regeneration.

We have emphasized the importance of chromatin regulation as an early step of the regeneration process in the introduction. Moreover, we have further discussed the importance of pioneer transcription factors as key players in this process.

2. It would be nice to see an exemplary figure panel of the ATAC and ChIPmentation data in Figure 1 (“seeing is believing”).

This is a very good suggestion. As mentioned, we have added an example plot for each condition (Figure S2 and S3) to better visualize and clarify how each mentioned accessible chromatin region behaves in *wnt1* and *notum* (RNAi) conditions. In this plot together with the genome browser, the H3K27ac ChIP-seq coverage tracks have been included to better understand the chromatin accessibility.

3. I did not see a description of how many replicates were used in ATAC or ChIP experiments, although in the methods it states that “Twenty animals were used per biological replicate.” Perhaps I just missed it, but please add $n = __$ in either the main text (preferably) or appropriate methods section, so the reader can adequately interpret the reproducibility of the data.

Thanks to the reviewer's comment. We have included a methods section, carefully explaining the number of animals used per biological replicate, and a detailed protocol for ATAC-seq and ChIP-seq experiments.

4. The label in Figure panel 2C is a mess...

Thank you for the reviewer's observation. Figure 2C has been improved. We think that now it is more understandable.

5. Figure 3a, the 'Reads coverage' data and ATAC-seq peaks don't seem to overlap for Enhancer 1, can this be explained?

We thank the reviewer's observation. Figure 3a has been improved and expanded in Figure S7. Now it also includes H3K27ac ChIP reads coverage and ATAC-seq MACS2 Narrow peaks. With this new plot, the reviewer can better observe how MACS use the two sources to predict the narrow peaks in enhancer 1.

6. In Figure 4b, what criteria was used for 'enhancer activity evidences'. Presumably some combination of chip/atac data, which is totally fine, but this should be explained in the methods section, and possibly figure legend. Also, 'evidences' should just be 'evidence'.

The enhancer activity research was performed manually, checking the distribution coverage of two main histone modifications associated with enhancer activity (i.e. H3K27ac, H3Kme1/2/3). When possible, both parameters were used. We have clarified this aspect in the manuscript and a new method section has been added describing how the analysis was performed.

We have also corrected the "evidence" word in the manuscript.

Of note, Figure 4 is now Figure 3b and c.

7. In figure 5a, 70% of foxG RNAi animals showed a tailless phenotype, 10% showed tails, and 20% showed... presumably two-headed?

In the mentioned Figure 5a, the reviewer can observe that 7/10 (now 15/20) animals show a tailless phenotype, and 1/10 (now 2/20) present a two-headed phenotype. The rest (i.e., 2/10) present a normal phenotype. We have added arrows to point to the eye spots in Figure 5a , and also in Figure 5c, which is a magnification of the same image.

8. I found the last results section "FoxG specifies posterior identity" a bit confusing at first, and think the manuscript would be strengthened if this section is re-written. This is a very exciting aspect of the work, which should be described as clearly as possible for full impact. First, I didn't understand on the first pass that these were cut and regenerated animals, which is a critical point of the experiment. Additionally:

"Eventually, some animals showed the strongest two-headed phenotype".

What does 'eventually' mean here, after some amount of time? If so, how much? How many animals exhibited this phenotype? Also, what does the 'strongest' two-headed phenotype mean, stronger than what, other phenotypes?

Moreover, there are references to Fig 4 that are clearly meant to be references to Fig 5.

We thank the reviewer for pointing these out. We have now split this section in two, with the aim to present the results more clearly. Furthermore, we have clarified that the phenotypes were observed in regenerating animals, and we have re-explained the inhibition protocol used (in two independent experiments) to obtain two-headed planarians.

We have also corrected the Figure annotation in the paragraph.

9. The Introduction could also benefit from an elaboration of early vs. late Wnt expression, as this is an important distinction for phenotypic outcomes in RNAi experiments and interpretations of results.

Thanks, we have clarified this aspect in the text.

10. Definitely not necessary for publication, but I think there are interesting connections to pioneer TFs and early developmental processes, and context could be included in the Intro. or discussion to broaden the interest of this work beyond planarian regeneration.

This interesting point has been briefly addressed now in the discussion section.

REVIEWERS' COMMENTS

Reviewer #1 (Remarks to the Author):

The authors have added some clarity to their analysis by providing a more complete explanation of their data sets which reveals that their data sets are predictably noisy, as is the case with many such studies. Although the FoxG work is very clear, it is still unclear how valuable their genome wide data is, and while their FoxG work is very compelling I note that they cite a paper that identifies enhancers genome wide in stem cells. While their samples are mixed tissue the most common cell type in them will be stem cells as this is the most numerous cell type in planarians. Would it not be useful to make broader comparisons with this data set to see if similar CREs are implicated?

The authors explain that missing RNA-seq timepoints are because the timepoints did not have met their prior expectations regarding Wnt pathway expression. Is this because RNAi did not work in these samples? I am not clear that it is correct to remove samples from an experiment because the experimental manipulation did not result in a priori expectations. Nonetheless it may not be practical redoing the timepoints if RNAi failed and was not checked independently (by phenotypic or other means) before RNA-seq. Can the authors be sure that RNAi at the other timepoints was also not hypomorphic?

Reviewer #2 (Remarks to the Author):

I appreciate the thorough additions and revisions to this manuscript. Given the extensive response to reviewers and new elements of the paper, I believe that it should be accepted and published in Nature Communications at this time. Congratulations to the authors on an exciting project and paper!

Reviewer #3 (Remarks to the Author):

Review comments:

1. I found it disappointing that the authors did not conduct a second replicate of t'0. No discussion was made to explain why they did not add one, although it seems well within their capabilities. Ultimately, it is not an absolute requirement for me, because their current data set is sufficient to make their main conclusions. But, it would have strengthened them, and added another dimension to their data set. I also note that all reviewers made a similar comment/request in the initial reviews.

2. The authors made a considerable effort to improve their methods sections (both wet-lab and bioinformatic analyses). Good. But, I am afraid I still have some problems with the description of their ATAC-seq analyses.

In the methods section (pg 28):

"Significant differentially bound sites obtained from ATAC-seq anterior versus posterior comparison, 611 and 2484 respectively, were crossmatched by DiffBind against the corresponding anterior and posterior ChIP-seq control samples (gfp-RNAi) [presumably for H3K27ac]. Such procedure returned 555 and 1869 non-significant sites that were marked as putative enhancers for anterior and posterior respectively; those enhancers were assigned to "accessible" state at this point and will be referred to as ATAC-ChIP regions."

I am confused why these are now considered non-significant? Perhaps even more confusingly, later they state that the 'significant' ATAC-peak sites that overlapped with ATAC peaks from the RNAi knockdown were then considered "slightly" or "less" accessible... Why?? What makes them 'less' accessible? Further, 'slightly' and 'less' are still not defined in the main text methods, although they remain categories in Figure 1. It pains me to say this, but I believe the authors still need to clarify the logic of these analyses for readers to properly interpret their findings.

3. The authors make no explanation for why their pie-charts were incorrect in the first version. Simple mistakes or the use of older versions of data are presumably the cause - which is fine - but without an explanation, it's a little disconcerting.

4. I was pleased with the conservation analysis of Intron 1 and enhancer elements in the updated figure 3.

Summary of review: I still think this manuscript is deserving of publication in Nature Comm. because of the novel data sets, earlier than expected changes to chromatin, and functional analysis of FoxG, but I still think there are a few hurdles the authors need to clear (especially point #2). But, of course, that is up to the editor.

REVIEWERS'

COMMENTS

Reviewer #1 (Remarks to the Author):

The authors have added some clarity to their analysis by providing a more complete explanation of their data sets which reveals that there data sets are predictably noisy, as is the case with many such studies. Although the FoxG work is very clear, it is still unclear how valuable there genome wide data is, and while their FoxG work is very compelling I note that they cite a paper that identifies enhancers genome wide in stem cells. While there samples are mixed tissue the most common cell type in them will be stem cells as this the most numerous cell type in planarians. Would it not be useful to make broader comparisons with this data set to see if similiary CREs are implicated?

Neoblasts are not the major cell type in planarians. In an intact worm the percentage is around 25%, according to Plass et al 2018. Furthermore, in the study to which we refer, Jakke Neuro et al. (2022) performed ATACseq of a subpopulation of cycling neoblasts, the X1 population, which is x-ray insensitive, and are at S/G2/M. Thus, the percentage of this X1 population in our samples could be less than 10%, which makes this comparison not very useful. As explained in the last revision, the important point of our study is to analyze an 'in vivo' sample, with complex cell types. The comparison with the data obtained in the 'in vitro' X1 population could be useful in further analysis when trying to analyze cell specific regulatory regions.

The authors explain that missing RNA-seq timepoints are because the timepoints did not have expect their prior expectations regarding Wnt pathway expression. Is this because RNAi did not work in these samples? I am not clear that it is correct to remove samples from an experiment because the experimental manipulation did not result in a priori expectations. Nonetheless it may not be practical redoing the timepoints if RNAi failed and was not checked independently (by phenotypic of other means) before RNA-seq. Can the authors be sure that RNAi at the other timepoints was also not hypomorphic?

When analyzing the differential gene expression for the first time on all RNA-Seq replicates (once discarded two libraries due to low sequencing depth with respect all the rest, as described in the new "Statistics and Reproducibility" section of the manuscript), we noticed that two *wnt1*-RNAi replicates, one at 0 hR and another at 12 hR, had expression levels of *wnt1* above the average expression level for those timepoints of controls. Therefore, we considered RNAi inactivation of *wnt1* was not working quite well on those replicates, and, because of that, it was better to discard those two replicates. As explained in the last revisions, our goal was to obtain tailles planarians and not bi-headed, and that is the reason why the levels of inhibition of *wnt1* were not very high. This is not failing of the protocol, but it is intrinsic to the protocol of inhibition we used, and considering that we could check the samples that showed *wnt1* levels downregulated. In all the other replicates *wnt1* expression levels from *wnt1*-RNAi samples were lower than those for the control samples, as expected (Figure CSM.10 panel on the left).

After re-running the differential expression analysis again once those two replicates were removed, this was neither affecting *wnt1* expression patterns through time nor the proper separation between control and *wnt1*-RNAi samples, as it can be appreciated on the right panel of Figure CSM.10.

We would like to thank the Reviewer for suggesting all the ATACseq and ChIPseq QC checkpoints from the previous round of revisions. We think that now the paper has substantially improved and its reproducibility is demonstrated.

Reviewer #2 (Remarks to the Author):

I appreciate the thorough additions and revisions to this manuscript. Given the extensive response to reviewers and new elements of the paper, I believe that it should be accepted and published in Nature Communications at this time. Congratulations to the authors on an exciting project and paper!

We thank the reviewer for all the comments and good points brought up, which helped us to definitely improve the manuscript.

Reviewer #3 (Remarks to the Author):

Review comments:

1. I found it disappointing that the authors did not conduct a second replicate of t'0. No discussion was made to explain why they did not add one, although it seems well within their capabilities. Ultimately, it is not an absolute requirement for me, because their current data set is sufficient to make their main conclusions. But, it would have strengthened them, and added another dimension to their data set. I also note that all reviewers made a similar comment/request in the initial reviews.

We acknowledge the point of view of the reviewer, and agree with the fact that having a second or third replicate at time 0 would have allowed us to explore CRES specifically accessible during regeneration. However, at the initial stage of the project the focus was to compare control and RNAi animals at 1 time point (12h), which was already very hard, since we analyzed wound regions, containing few cells.

2. The authors made a considerable effort to improve their methods sections (both wet-lab and bioinformatic analyses). Good. But, I am afraid I still have some problems with the description of their ATAC-seq analyses.

In the methods section (pg 28):

"Significant differentially bound sites obtained from ATAC-seq anterior versus posterior comparison, 611 and 2484 respectively, were crossmatched by DiffBind against the corresponding anterior and posterior ChIP-seq control samples (gfp-RNAi) [presumably for H3K27ac]. Such procedure returned 555 and 1869 non-significant sites that were marked as putative enhancers for anterior and posterior respectively; those enhancers were assigned to "accessible" state at this point and will be referred to as ATAC-ChIP regions."

I am confused why these are now considered non-significant? Perhaps even more confusingly, later they state that the 'significant' ATAC-peak sites that overlapped with ATAC peaks from the RNAi knockdown were then considered "slightly" or "less" accessible... Why?? What makes them 'less' accessible? Further, 'slightly' and 'less' are still not defined in the main text methods, although they remain categories in Figure 1. It pains me to say this, but I believe the authors still need to clarify the logic of these analyses for readers to properly interpret their findings.

The reviewer is right. Thanks. We have clarified the paragraph. We think that with the new information added, now it is clear how we named all the different conditions. We hope that the reviewer finds it clear.

3. The authors make no explanation for why their pie-charts were incorrect in the first version. Simple mistakes or the use of older versions of data are presumably the cause - which is fine - but without an explanation, it's a little disconcerting.

The reviewer is absolutely right. Another reviewer noticed in the previous revision round, and we addressed it. We used an old data set on that first version.

4. I was pleased with the conservation analysis of Intron 1 and enhancer elements in the updated figure 3.

We thank the reviewer for all the good suggestions regarding the Intron 1 and enhancer elements that help us to improve the manuscript and the figures.

Summary of review: I still think this manuscript is deserving of publication in Nature Comm. because of the novel data sets, earlier than expected changes to chromatin, and functional analysis of FoxG, but I still think there are a few hurdles the authors need to clear (especially point #2). But, of course, that is up to the editor.

We would like to thank this reviewer for all the great suggestions and comments that were brought up during this review process.